# Estradiol Modulates the Sensitivity to Vancomycin of *Lactobacillus paracasei* and *Staphylococcus aureus* Biofilms—Constituents of Human Skin and Vaginal Microbiota

**DOI:** 10.3390/microorganisms13122777

**Published:** 2025-12-05

**Authors:** Anna M. Mosolova, Nadezhda A. Loginova, Ecaterina V. Diuvenji, Artem G. Chebotarevskii, Marina V. Sukhacheva, Sergey V. Tsibulnikov, Polina Y. Bikmulina, Vera M. Tereshina, Elena A. Ianutsevich, Olga A. Danilova, Aleksandra S. Novikova, Vladimir K. Plakunov, Sergey V. Martyanov, Alexander I. Netrusov, Andrei V. Gannesen

**Affiliations:** 1Federal Research Center ‘Fundamentals of Biotechnology’ of Russian Academy of Sciences, Moscow 117312, Russia; a.mosolova@fbras.ru (A.M.M.); n.loginova@fbras.ru (N.A.L.); e.duvenzhi@fbras.ru (E.V.D.); a.chebotarevsky@fbras.ru (A.G.C.); sukhacheva@biengi.ac.ru (M.V.S.); v.tereshina@fbras.ru (V.M.T.); e.yanutsevich@fbras.ru (E.A.I.); o.danilova@fbras.ru (O.A.D.); v.plakunov@fbras.ru (V.K.P.); s.martyanov@fbras.ru (S.V.M.); 2Institute for Regenerative Medicine, Sechenov University, Moscow 119991, Russia; tsibulnikov_s_v@staff.sechenov.ru (S.V.T.); bikmuina_p_yu@staff.sechenov.ru (P.Y.B.); 3LLC ‘BAVAR+’, Moscow 129626, Russia; novikova@bavarswiss.ru; 4Microbiology Department, Lomonosov Moscow State University, Moscow 119234, Russia

**Keywords:** biofilms, hormones, estradiol, microbial endocrinology, antibiotics, vancomycin, antibiotic resistance, *Lactobacillus*, *Staphylococcus aureus*

## Abstract

We investigated the effects of vancomycin, estradiol, ethanol, and their combinations on the growth of mono- and binary-species biofilms of Lactobacillus paracasei and Staphylococcus aureus. It was found that vancomycin at a subinhibitory concentration of 0.001 µg/mL, estradiol, and ethanol acted antagonistically in all cases. This effect was observed across all strains studied. Furthermore, the effects of the active compounds were evident at population, cellular and molecular levels, and were reflected in changes to the count of colony-forming units (CFUs), gene expression, and the physiological and biochemical characteristics of cells (e.g., lipid composition of membranes and the extracellular matrix). Therefore, at subinhibitory concentrations of vancomycin in the medium, estradiol can modulate the antibiotic’s effect on biofilms, thereby regulating deeply microbial communities.

## 1. Introduction

The human microbiota is tightly interconnected with the human organism through signalling and metabolic exchange [1]. Today, aspects of this interkingdom signalling and the regulatory pathways of the holobiont ‘human and microbiota’ [2] are studied within the framework of microbial endocrinology [3]. The human microbiota comprises thousands of species from all three domains of life, as well as a myriad of viruses [4], and predominantly exists in the form of biofilms [5]. Biofilms are highly complex microbial communities; even monospecies models pose challenges in medical, industrial and microbiological respects [6,7]. Hence, the regulatory processes within microbial biofilms require extensive investigation.

A prospective strategy for combating biofilm infections is to identify enhancers for existing drugs and antibiotics, thereby making them more effective against biofilms [8]. Those enhancers are compounds that lack direct antimicrobial activity, which reduces the likelihood of microorganisms developing resistance. Such compounds may be produced naturally by the host and include humoral factors, such as hormones and neurotransmitters. Additionally, these regulatory compounds can modulate the microbiota, shifting the balance toward commensal microbes and enhancing their protective properties. At the beginning of 21st century, several studies demonstrated the impact of certain hormones on bacteria’s susceptibility to antibiotics [9,10]. In recent years, additional data has emerged [11,12].

The effects of hormones on microorganisms are well-known. For example, catecholamines such as norepinephrine and epinephrine significantly stimulate the growth of various bacteria, and mechanisms of hormone action have been proposed for some members of the *Enterobacteriaceae* family [13]. Natriuretic peptides regulate the metabolism and molecular state of multispecies biofilms of *Cutibacterium acnes* and *Staphylococcus aureus* or *Staphylococcus epidermidis* [14,15]. A recent review provides a comprehensive overview of hormones as microbial effectors [16]. Steroids, particularly estradiol, have also been a focus of recent research. Estradiol has been shown to affect *Lactobacillus crispatus* [17,18], *Pseudomonas aeruginosa* [19], and various soil microorganisms [20]. For example, estradiol and other steroid hormones can modulate quorum-sensing systems in *P. aeruginosa* [21,22]. The growth and virulence of *Prevotella aurantica*, a pathogen isolated from the human oral cavity, are stimulated by estradiol via the dipeptidyl peptidase IV pathway [23]. Additionally, estradiol alters the virulence of uropathogenic *E. coli* [24]. There are advances in the understanding of the effects of steroids on bacteria, microbial steroid metabolism and steroid-induced alterations in microbiota structure [25]. However, while estradiol’s influence on Gram-negative pathogens is relatively well documented, its role in Gram-positive biofilms and antibiotic modulation, and the mechanisms of estradiol action on microorganisms and their variability remain poorly understood. For example, the mechanism by which estradiol shifts *S. aureus* adherence to epithelial cells via the estrogen receptor α-associated signaling pathway in eukaryotic cells is now knowns [26]. However, the processes occurring in bacterial cells of different microbial species are not well understood. Therefore, it is crucial to investigate how human steroid hormones influence other members of the microbiota within biofilms, and how they modulate the efficacy of antibiotics. We started from 17β-estradiol, a hormone frequently used in pharmacology as a component of hormonal therapies, contraceptive etc.

*Lactobacillus* species are typically the dominant constituents in the human vaginal microbiota and help maintain a healthy microenvironment by lowering pH, producing bacteriocins, and generating hydrogen peroxide (H_2_O_2_) [27]. Specifically, *Lactobacillus paracasei* [28] and *Lactobacillus gasseri* [29] are components of healthy vaginal microbiota. However, *S. aureus* can colonize the vaginal mucosa and cause disorders such as aerobic vaginitis. The frequency of *S. aureus* vaginal colonization can be high, reaching up to 40% in certain groups [30]. Due to the extreme toughness of some *S. aureus* strains, such as MRSA (methicillin-resistant *S. aureus*), specific drugs are needed to treeat MRSA and *S. aureus* in general. Vancomycin is one such antibiotic used to treat *S. aureus*-associated vaginosis [31]. Since these bacteria colonize the vaginal mucosa and are exposed to estradiol and vancomycin, it is important to investigate the potential combined effects. Due to the strong antibacterial properties of ethanol, which is produced by various vaginal microorganisms [32] and used as an estradiol solvent, it is important to also check the combined effect of ethanol and vancomycin as an additional control. The aim of this study is therefore to examine how estradiol modulates vancomycin activity against mono- and dual-species biofilms of *S. aureus* and *L. paracasei*, including *L. gasseri* as an additional microbial marker. We hypothesize that estradiol actually is able to modify efficacy of vancomycin and biofilm development in mono- and dual-species systems of *L. paracasei* and *S. aureus*.

## 2. Materials and Methods

### 2.1. Strains and Storage

*Staphylococcus aureus* 209P (ATCC 6538P) is a commonly used strain for investigating the effects of antibiotics and is maintained in many microbial collections, including the VKM collection (Pushchino, Russia). *Lactobacillus paracasei* AK508 was obtained from the Lomonosov Moscow State University collection (Moscow, Russia). The AK508 strain was originally isolated from kvass wort. *L. paracasei* 27W was obtained from the Russian Collection of Industrial Microorganisms (NRC “Kurchatov Institute”, Moscow, Russia). *Lactobacillus gasseri* ATCC 33323, a type of strain that was isolated from vaginal mucus, was obtained from the Korean Collection for Type Cultures (KCTC, Jeongeup, South Korea). All strains were stored in liquid nitrogen at −196 °C.

### 2.2. Active Compounds

Vancomycin (FarmConcept, Redkino, Russia) was supplied as a sterile lyophilized powder. It was dissolved in 10 mL of sterile Milli-Q water and stored at −20 °C for up to 21 days. For the experiments, serial dilutions were prepared in a sterile medium to achieve the following test concentrations: 0.001, 0.01, 0.05, 0.1, 0.5, 1, 2, 4, 6, 8, 10, 20, 30, 40, 100, and 500 µg/mL.

β-Estradiol (Merck, Darmstadt, Germany) was dissolved in ethanol (Donskoy Distillery, Yepifan, Russia) to achieve a final stock concentration of 100 mg/mL, then stored at −20 °C. Also, estradiol stock solutions were kept in the dark to prevent potential light-mediated degradation. For the experiments, estradiol stock solutions in ethanol were prepared and added to a liquid or 1.5% agar medium at volumes yielding a baseline concentration of 2.2 × 10^−10^ M in the medium, as previously described [33]. This concentration corresponds to the normal physiological level in the blood plasma of midluteal-phase women [34]. Blood plasma concentration is a more stable parameter because the concentration of estradiol in vaginal mucus is highly variable. It depends not only on the phase of the menstrual cycle, but also on vaginal health and the use of hormonal drugs. eSince ethanol, the solvent for estradiol, can affect bacterial cells, equivalent volumes of ethanol were added to the corresponding control media. The final ethanol concentration in the medium was 0.06‰.

### 2.3. Bacterial Cultures

*Staphylococcus aureus* 209P, stored in liquid nitrogen, was streaked into tubes containing 5 mL of semi-solid lysogeny broth (LB, Dia-M, Moscow, Russia), supplemented with 0.5% agar (Dia-M, Moscow, Russia). Tubes The tubes were incubated and then stored as described previously [15,33]. Liquid cultures of *S. aureus* 209P were obtained and adjusted as described previously [15]. 24 h cultures were obtained, and their OD_540_ was adjusted as described previously [15]. Cultures of *Lactobacillus paracasei* AK508 were prepared as described previously [33].

*Lactobacillus gasseri* ATCC 33323 was cultivated similarly to *L. paracasei* AK508, except for preculture on modified de Man–Rogosa–Sharpe (MRS) medium with the following composition (g/L): peptone (Dia-M) 20; glucose (Dia-M) 20; yeast extract (Dia-M) 5; sodium acetate (Reachem) 5; ammonium chloride (Reachem) 2; sodium citrate (Reachem) 2; potassium diphosphate (Reachem) 2; Tween 80 (Dia-M) 1; magnesium sulfate (Reachem) 0.1; manganese sulfate (Reachem) 0.05; pH 7.0. Liquid MRS cultures were obtained after 72 h of incubation at 37 °C.

### 2.4. Cultivation of Monospecies and Dual-Species Biofilms

#### 2.4.1. Monospecies Biofilms on Polytetrafluoroethylene Cubes

Biofilms were grown anaerobically on polytetrafluoroethylene (PTFE) cubes for 72 h at 33 °C with shaking at 150 rpm as described previously [33]. As previously described, ethanol-only controls matched the estradiol solvent volume. An additional control was tested: a vancomycin plus ethanol combination. For the ethanol and vancomycin samples, tubes with no additions served as controls. For estradiol and vancomycin plus ethanol samples, ethanol-treated bacteria served as controls. For vancomycin plus estradiol samples, the vancomycin plus ethanol samples served as controls.

Biofilm fixation, CV-staining and OD_590_ of CV extracts in ethanol were measured as described previously [33].

Based on these results, a vancomycin concentration of 0.001 µg/mL, which produced a moderate and statistically significant effect on biofilms of at least one type of biofilm, was selected for subsequent experiments.

#### 2.4.2. Monospecies and Dual-Species Biofilms on Glass Fiber Filters

For counting colony-forming units (CFUs), analyzing metabolic activity, and extracting RNA, biofilms were obtained on the surface of GF/F glass microfiber filters (Whatman, Maidstone, UK). Bacterial cultures were prepared as previously described [15,33]. Two inoculation schemes were used. First, the suspensions were adjusted to an OD_540_ of 1 with PS, and then they were mixed in equal volumes to obtain dual-species cultures, as previously described [33,35]. Monospecies cultures for this scheme were prepared by diluting an OD 1 suspension 1:1 with saline. Second, since lactobacilli markedly inhibited staphylococci, an additional series used different initial ODs: *S. aureus* was adjusted to an OD_540_ of 2 and *L. paracasei* was adjusted to an OD_540_ of 0.5. Dual-species cultures were obtained by mixing these suspensions in equal volumes.

Biofilms were formed on 1.5% RCM agar supplemented with active compounds (2.2 × 10^−10^ M estradiol, 0.06% ethanol, and 0.001 µg/mL vancomycin) or their combinations. The compounds were added to sterile glass vials, followed by 20 mL of melted RCM agar cooled to 50 °C. The mixture was then poured into Petri dishes. Sterile Whatman GF/F glass fiber filters (20 × 20 mm; nominal pore diameter 200 nm) were placed on the solidified medium. Ten microliters of the prepared mono- or dual-species culture were applied to the center of each filter; and each condition was duplicated on two filters. The biofilms were then incubated anaerobically for 72 h at 33 °C.

#### 2.4.3. Monospecies and Dual-Species Biofilms in 24-Well Glass-Bottom Plates

For confocal microscopy, biofilms were grown in 24-well flat-bottom glass plates (Eppendorf, Hamburg, Germany) for 72 h at 33 °C. The inocula were prepared as described in Section 2.4.2. *L. paracasei* was adjusted to an OD_540_ of 0.5 and *S. aureus* to an OD_540_ of 2. Dual-species communities were obtained by mixing the monospecies suspensions in equal volumes. Then, one milliliter of liquid RCM, with or without active compounds at the indicated concentrations, and 17 µL of the prepared cell suspension were added to each well. The plates were then incubated anaerobically for 72 h using a GasPak bags (BD, Franklin Lakes, NJ, USA) supplemented with Anaerogaz sashets (NIKI-MLT, Saint Petersburg, Russia). Anaerogas sachets provide 14–16% of CO_2_, 4–6% of H_2_ and <0.1% of O_2_. After incubation, the supernatant was removed, and the biofilms were washed once with sterile PS. The PS was discarded and plates were air-dried at RT. The biofilms were fixed with 1 mL of 96% ethanol for 20 min at RT. The ethanol was then discarded, and plates were dried at RT. The fixed samples were stained by fluorescent in situ hybridization (FISH).

#### 2.4.4. Biofilms in 96-Well Polystyrene Microtiter Plates

To evaluate the impact of estradiol on bacteria when absorbed onto a carrier surface, biofilms were cultivated in 96-well microtiter plates. The 96-well polystyrene plates (Wuxi Nest Biotechnology, Wuxi, China) were chosen for high-throughput adhesion test because their surface properties are similar to those of the PTFE. Estradiol was diluted to 2.2 × 10^−10^ M in three solvents: sterile physiological saline (PS), ethanol, or sterile RCM. For the wells that were to receive estradiol, 200 µL of the estradiol solution in one of the solvents was added. Control wells received 200 µL of the corresponding solvent without estradiol. The plates were sealed with Parafilm (Amcor, Zurich, Switzerland) and incubated overnight at 33 °C for estradiol adsorption. The liquids were then removed, and the wells were washed once with sterile PS. Then, 200 µL of liquid RCM with or without test additions (excluding estradiol) was added to each well.

Next, 3.3 µL of the prepared bacterial suspension was inoculated per well. Monospecies suspensions were adjusted to an OD_540_ of 0.5. The dual-species suspension was a 1:1 mixture of suspensions each with an OD_540_ of 1. The plates were then incubated anaerobically for 72 h using the GasPak-Anaerogaz system as previously described, and processed using the standard crystal violet protocol [36].

To compare with surface-adsorbed estradiol, we evaluated the standard mode of estradiol addition to the medium. In microtiter plates, 200 µL per well of RCM without additions or with 0.06% ethanol, 2.2 × 10^−10^ M estradiol, 0.001 µg/mL vancomycin, or their combinations was dispensed. Then 3.3 µL of bacterial suspension was added. Plates were incubated anaerobically and aerobically for 72 h at 33 °C. Planktonic and biofilm measurements were carried out as above. To assess generality, we also examined *L. gasseri* ATCC 33323 monospecies biofilms and *L. gasseri*–*S. aureus* dual-species biofilms.

#### 2.4.5. Biofilms on Cellulose Filters

Because of the ability of estradiol to affect cell membranes [19], we analyzed the lipid composition of cells and biofilm matrix of *L. paracasei*. For that, biofilms were obtained on the surface of cellulose filters. Two strains were studied: the kvass-derived AK508 and the gut-derived 27 W. Biofilms were grown as described previously [37] with modifications. Briefly, lactobacilli were grown on 1.5% MRS agar (for higher biomass yield) in the presence of ethanol, 2.2 × 10^−10^ M estradiol, 2.2 × 10^−8^ M estradiol, or without additions. The higher estradiol concentration was included based on prior observations [33]. A sterile cellulose filter (Ozon, Russia) was placed on the agar surface, and 500 µL of culture adjusted to OD_540_ = 0.5 was spread onto the filter. Biofilms were incubated anaerobically for 72 h at 33 °C.

### 2.5. Staining of Biofilms with Crystal Violet

#### 2.5.1. Biofilms on the PTFE Cubes

After incubation, OD_540_ of planktonic cultures was measured in 1 cm pathlength cuvettes using the corresponding blank. The remaining cell suspension was removed, cubes were rinsed twice with tap water, and biofilms were fixed by adding 3 mL of 96% ethanol per tube for 15 min at RT. Ethanol was removed, and 3 mL of 0.1% crystal violet (CV, Merck, Darmstadt, Germany) solution was added per tube for 20 min at RT. Tubes were rinsed six times with tap water until the wash was colorless. Cubes were blotted with a paper towel, transferred to clean tubes, and 3 mL of ethanol was added to extract the dye. Tubes were capped and stored at RT for 24 h. OD_590_ of CV extracts was then measured in 0.5 cm pathlength cuvettes using the appropriate blank.

#### 2.5.2. Biofilms in 96-Well Microtiter Plates

Briefly, planktonic culture was transferred to a new plate for OD_540_ measurement. Wells were washed once with sterile PS, blotted, air-dried, and fixed with 200 µL ethanol per well for 20 min at room temperature. Ethanol was removed, plates were blotted and air-dried, then stained with 0.1% crystal violet (200 µL per well) for 20 min at room temperature. Stain was removed, wells were washed six times with tap water, and air-dried. Ethanol (200 µL per well) was added to extract the bound dye for at least 2 h at room temperature. OD_590_ of the extracts was then measured.

All measurements were performed using an XMark microplate spectrophotometer (Bio-Rad Laboratories, Hercules, CA, USA). Controls matched those in Section 2.5.

### 2.6. Counting of Colony-Forming Units in Biofilms on the Glass Fiber Filters

After incubation, a filter with biofilms was dispersed in sterile PS in glass tubes, as described previously [33]. The resulting suspension was serially diluted tenfold in PS and plated on RCM agar to count colony-forming units (CFU). To enumerate *L. paracasei* CFU in dual-species communities, vancomycin (4 µg/mL) was added to RCM to selectively inhibit *S. aureus*. Colonies were counted after 72 h of incubation.

### 2.7. Assessment of Metabolic Activity in Biofilms on the Glass Fiber Filters

After incubation, filters with biofilms were transferred to a 6-well culture plate. Three milliliters of 0.1% MTT (4,5-dimethylthiazol-2-yl-2,5-diphenyltetrazolium bromide) in sterile LB was added per well, and biofilms were incubated for 30 min at RT. Filters were washed with distilled water to remove residual dye, and the insoluble formazan was extracted in 3 mL of dimethyl sulfoxide (DMSO, Ekos-1, Moscow, Russia) for at least 24 h. Metabolic activity was assessed by measuring OD_540_ of the formazan extracts, as described previously [33].

### 2.8. Fluorescent in Situ Hybridization

Fluorescent in situ hybridization (FISH) was performed as described previously [15,33,38]. Target-specific probes and fluorophores are listed in Table 1. Probes and fluorophores were obtained from Syntol (Moscow, Russia). The absence of false-positive reactions of probes was validated as described previously [15] and in references [39,40].

Fixed biofilms in wells were pretreated for 15 min with lysozyme (1 mg/mL; Merck, Darmstadt, Germany) in 10 mM Tris-HCl buffer, pH 8.0, followed by lysostaphin (10 µg/mL; Merck) in 10 mM Tris-HCl, pH 8.0, for 5 min. To increase cell membrane permeability, biofilms were sequentially treated at room temperature with ethanol in phosphate buffer (pH 7.4) at 50%, 80%, and 96% for 3 min each. After the final 96% ethanol step, plates were air-dried and hybridization was performed.

For *S. aureus*, 200 µL of hybridization buffer was added per well (0.9 M NaCl, 20 mM Tris-HCl, 0.01% SDS, 20% formamide, pH 8.0). All buffer components were from Merck (Darmstadt, Germany) except NaCl (Dia-M, Moscow, Russia). The buffer contained 10 pmol of the *S. aureus* probe. Plates were incubated for 90 min at 46 °C in a humid chamber. The hybridization buffer was then removed and 200 µL of wash buffer (225 mM NaCl, 20 mM Tris-HCl, 0.01% SDS) was added per well for 15 min at 48 °C.

For *L. paracasei*, 200 µL of hybridization buffer was added per well (20% formamide, 0.9 M NaCl, 20 mM Tris-HCl, 0.02% SDS) containing 10 pmol of the *Lactobacillus* probe. Hybridization was carried out in a humid chamber at 48 °C for 60 min. The buffer was discarded and 200 µL of wash buffer (0.9 M NaCl, 20 mM Tris-HCl) was added per well, followed by incubation in a humid chamber at 50 °C for 15 min.

After washing and air-drying, a drop of ProLong Gold Antifade Mountant with DAPI (Thermo Fisher Scientific, Waltham, MA, USA) was added to each well. Plates were wrapped in aluminum foil and stored at 4 °C for at least 24 h.

### 2.9. Confocal Microscopy of Biofilms

Confocal laser scanning microscopy was used to analyze biofilm architecture and the effects of active compounds on species interactions in dual-species biofilms of *L. paracasei* and *S. aureus*. Monospecies biofilms were analyzed in parallel.

Biofilms stained with FISH probes were imaged on an Olympus (Olympus, Hachioji, Tokyo, Japan) IX83P2ZF confocal microscope with a UPlanSApo (Olympus, Hachioji, Tokyo, Japan) 60×/1.42 oil immersion objective. Diode lasers at 481 and 561 nm were used for fluorescein amidite (FAM) and rhodamine 6G (R6G), respectively. Final image resolution was 512 × 515 pixels. For dual-species biofilms, half of the samples were labeled with a probe specific for *S. aureus* and the other half with a probe specific for *L. paracasei* (see below). Laser power was set to 10% of maximum.

Three-dimensional imaging and blinded analysis were performed as described previously [33] with a z-step of 0.36 µm. OIB files were processed in ImageJ 1.48v Java 1.6.0_20 (NIH, Bethesda, MD, USA) using the Bio-Formats importer and the Comstat2 plugin (Version 2.1.1 July 2015, University of Copenhagen, Copenhagen, Denmark). Data were converted to uncompressed OME-TIFF and saved as single files. For each OME-TIFF, the “biomass” parameter was calculated in Comstat2 (Version 2.1.1) with an individual threshold of 1 and no volume filtering. At least five representative fields of view per sample were selected for analysis and used to construct three-dimensional images.

### 2.10. Antibacterial Activity Assay of S. aureus 209P and L. paracasei AK508

The, standard agar block method was applied as previously described [41]. The following test cultures were selected: *Staphylococcus epidermidis* ATCC 14990, *Micrococcus luteus* C01, *M. luteus* HB, *Pseudomonas. chlororaphis* 449, *Kytococcus schroeteri* H01, *Escherichia coli* K12, *Candida* spp.

Briefly, 30 mL of agar RCM with or without active compounds was added to a Petri dish. Then, and 100 µL of the *L. paracasei* culture (OD_540_ = 1) was plated with a spatula for lawn growth. The plates were then incubated anaerobically for 72 h at 33.5 °C. After, a total of 100 µL of night cultures of test bacteria in LB medium with an OD_540_ of 0.1 were inoculated into 20 mL of cooled RCM agar. The resulting mixture was plated onto Petri dishes for deep-plate growth. Then, 10 mm diameter agar blocks with a 72 h *L. paracasei* lawn were cut and placed on the agar surface with the test bacteria. The plates were then incubated for 24 h at 33.5 °C. Finally, the inhibition zone diameters were measured.

### 2.11. Isolation of L. paracasei Biofilm Matrix

The isolation of biofilm matrix was performed as described previously [37,42] with modifications (reducing of sonication time to 1 min). For more details, See the Appendix A.

### 2.12. Isolation and Analysis of L. paracasei Lipids

A sample of freeze-dried biomass was homogenized in isopropanol (Rushim, Russia) and incubated at 70 °C for 30 min [43]. After decanting the supernatant, the pellet was extracted twice at 70 °C with an isopropanol–chloroform mixture (1:1), and once with a 1:2 mixture. The total final extract was then evaporated using a rotary evaporator. The residue was dissolved in 9 mL of a 1:1 chloroform–methanol (Rushim, Russia) mixture with the addition of 12 mL of a 2.5% NaCl solution to remove water-soluble compounds. After phase separation, the chloroform layer was dried by passing through water-free sodium sulfate. Then, it was evaporated, and desiccated with a vacuum pump. The residue was dissolved in a chloroform–methanol mixture (2:1) and stored at −21 °C.

To extract lipids from the matrix, 20 mL of matrix suspension was mixed with 20 mL of the chloroform–methanol mixture (2:1). The mixture was vortexed for 30 s and left at room temperature for 30 min for phase separation. After the aqueous phase was extracted twice with 10 mL of chloroform under the same conditions, the lower chloroform phase was isolated. Then, 50 mL of 2.5% NaCl was added to the total extract. The mixture was kept overnight at 4 °C for to remove water-soluble compounds and allow for phase separation. Then, the chloroform layer was dried by passing through water-free sodium sulfate, evaporated, and desiccated with a vacuum pump. The residue was dissolved in the chloroform–methanol mixture (2:1) and stored at −21 °C.

Polar and neutral lipid analysis was performed as previously described [44,45,46]. For more details, please refer to Appendix A. Quantitative analysis was carried out using densitometry with Dens software, version 5.1.0.2 (Lenchrom, Saint Petersburg, Russia), in the linear approximation mode.

### 2.13. qPCR for Differential Expression of S. aureus Resistance Genes in Biofilms

To investigate changes in expression of antibiotic resistance genes, quantitative polymerase chain reaction (qPCR) was performed. Previous studies have shown that hormones, antibiotics, and their combinations can alter the expression of a wide range of resistance genes [12,47]. Thus, we analyzed not only glycopeptide resistance genes but also genes conferring resistance to other antibiotic classes. We used the The Comprehensive Antibiotic Resistance Database [48] to identify resistance genes in *S. aureus* 209P, a more dangerous bacterium. The genes found in the strain are described in Appendix A.

Total RNA was extracted from three independent experiments as described above. First-strand cDNA was synthesized using Moloney murine leukemia virus reverse transcriptase (Evrogen, Moscow, Russia) according to the manufacturer’s protocol. Specific primers (see Appendix A) were designed using the built-in Primer3 module for qPCR in Unipro UGENE v38.1 [49]. The primers were checked *in silico* [50]. To select optimal primer pairs, a single-hybridization test was performed with *S. aureus* total DNA prior to qPCR. No PCR products were detected for *L. paracasei* DNA, indicating no false positive results. The primers sequences are presented in the Appendix A. Total DNA was extracted from 24-h suspension cultures using the HiPure Bacterial DNA Kit (Magen, Guangzhou, China). To improve cell wall disruption, the pellets were frozen in liquid nitrogen and milled with glass as described for RNA extraction.

RNA isolation followed Ovcharova et al. with minor changes [38]. For detailed information please see the Appendix A.

qPCR was performed using PB PCR buffer (Syntol, Moscow, Russia) with SYBR Green I and ROX passive reference dye (Syntol) for signal normalization. In each experiment, each sample was run in duplicate, and ddH_2_O (Syntol) served as the negative control. Amplification was carried out using the following program on a CFX96 Touch real-time PCR detection system (Bio-Rad, Hercules, CA, USA): polymerase activation 5 min at 95 °C, followed by 40 cycles of 15 s at 95 °C, 20 s at 55 °C, and 40 s at 62 °C. Differential expression was calculated relative to the untreated control. Target gene Ct means were normalized to the *S. aureus* 16S rRNA reference (accession NR_118997.2). Relative quantities were computed using the 2^−ΔΔCt^ method. The data were analyzed using CFX Manager v1.6.

### 2.14. Statistics

All experiments were performed at least three times independently with two or three technical replicates in each biological replicate. The data were analyzed using GraphPad Prism 8.3.0 (GraphPad Software, Boston, MA, USA). Results are presented as bar graphs with means and standard error of the mean or standard deviation, as indicated. The nonparametric Mann–Whitney U-test was used to assess statistical significance between groups in microbiological experiments. A Student’s *t*-test was used to assess statistical significance in gene expression experiments. Both tests were included in GraphPad Software (version 8.3.0 (538)).

## 3. Results

### 3.1. Screening of Vancomycin Effects on Monospecies Planktonic Cultures and Biofilms on PTFE Cubes

As expected, the biofilms of both bacteria were more resistant to vancomycin than planktonic cells. *L. paracasei* AK508 exhibited resistance: in planktonic culture, only 22% inhibition was observed at a concentration of 500 µg/mL of vancomycin in RCM. Meanwhile, slight but statistically significant inhibition of planktonic lactobacilli by 10–14% was observed at several concentrations without a linear trend (Figure 1A). Biofilms of *L. paracasei* AK508 were slightly stimulated by vancomycin; however 0.01 µg/mL of the antibiotic produced 31% inhibition (Figure 1C). Thus, vancomycin’s effect on lactobacilli was nonlinear, and planktonic growth inhibition appears to result from potential switching between planktonic and biofilm growth modes in this system.

For *S. aureus* 209P, both planktonic cultures and biofilms were strongly inhibited by vancomycin at concentrations of 2 µg/mL and above. At lower concentrations, planktonic cultures were stimulated, reaching a maximum of 18% at 1 µg/mL, whereas biofilms showed no linear dependence similarly to planktonic cultures. Notably, at 0.001 µg/mL, there was a significant 17% decrease in OD of CV extracts. This concentration is much lower than that used in clinical practice [51]; due to this interesting phenomenon, we decided to use this ultralow concentration of vancomycin. Additionally, because *S. aureus* is an opportunistic pathogen of higher clinical concern, we chose a vancomycin concentration effective against it. Therefore, we selected 0.001 µg/mL for subsequent experiments.

### 3.2. Estradiol Alters Vancomycin Effects on L. paracasei AK508 and S. aureus 209P in the PTFE-Cube System

We tested the combination of 0.001 µg/mL vancomycin and 2.2 × 10^−10^ M estradiol under anaerobic conditions. Estradiol enhanced the inhibition of *L. paracasei* biofilms by vancomycin (Figure 2E): biofilm biomass decreased by 21% relative to the ‘vancomycin plus ethanol’ control. Planktonic growth of lactobacilli was unaffected by any compound or combination.

Ethanol reduced planktonic growth of *S. aureus* by 11% and did not affect biofilms (Figure 2B). The combination of vancomycin and ethanol neutralized the antibiotic’s inhibitory effect on biofilms (Figure 2D). The addition of estradiol to vancomycin stimulated *S. aureus* biofilms by 13% (Figure 2F). Thus, in this system estradiol and ethanol acted additively to counteract the inhibitory effect of the antibiotic- and switch the response to stimulation. No analogous effects were observed in planktonic cultures, indicating higher susceptibility in the biofilm state.

### 3.3. Estradiol Modulates Vancomycin Responses in S. aureus 209P and L. paracasei AK508 Biofilms on Glass Fiber Filters

We used two inoculum levels for *S. aureus* 209P to quantify CFU and metabolic activity. For series 1, the suspensions were adjusted to an OD_540_ of 0.5, and for series 2, the suspensions were adjusted toto an OD_540_ of 2.0. *L. paracasei* was kept at an OD_540_ of 0.5 in both series. When inocula were of a standard OD_540_ of 0.5, dual-species biofilms contained a low number of *S. aureus* cells (only up to 1.2 × 10^6^ CFU per biofilm; see Figure 3A). Therefore, we increased the *S. aureus* inoculum to improve survival in the presence of lactobacilli. The effects of the treatments were more pronounced at a higher *S. aureus* OD (Figure 3B), especially in dual-species communities. This is a logical consequence of the shortage of *S. aureus* biomass that occurred during the plating of lower OD cultures. Lactobacilli suppressed *S. aureus* and no pronounced impact was detected. However, when plated in a higher amount, staphylococci were more resilient to the negative effects of *L. paracasei* and formed a more pronounced dual-species community.

Ethanol did not significantly affect the number of CFUs in either monospecies or dual-species biofilms. Vancomycin decreased the number of CFU in dual-species biofilms but slightly increased it in monospecies *S. aureus* biofilms. These effects were not statistically significant. However, ethanol enhanced the inhibitory action of vancomycin in dual-species biofilms. The number of *S. aureus* CFUs decreased from 3.9 × 10^9^ per biofilm in ethanol-treated samples to 1.0 × 10^9^ in ‘ethanol plus vancomycin’ samples, and from 1.7 × 10^9^ in 0.001 µg/mL vancomycin samples to 1.0 × 10^9^ in ethanol plus vancomycin. The number of *L. paracasei* CFUs decreased from 5.4 × 10^9^ in ethanol-treated samples to 3.1 × 10^9^ in ‘ethanol plus vancomycin’ samples; the latter matched the CFUs in vancomycin-treated biofilms (Figure 3B). Importantly, estradiol reduced the inhibitory effect of ethanol and vancomycin combination. This increased the number of CFU for both species in dual-species biofilms to 4.6 × 10^9^ (*S. aureus*) and 5.7 × 10^9^ (*L. paracasei*) in estradiol plus vancomycin samples. Thus, estradiol can modulate vancomycin efficacy.

By contrast, the effects on biofilm metabolic activity were modest, particularly when comparing ethanol plus vancomycin with estradiol plus vancomycin (Figure 3C,D). These results suggest that ethanol, vancomycin, and estradiol may primarily influence matrix properties and cell aggregation, altering aggregate size and retention rather than per-cell metabolism.

In summary, Table 2 shows the effects of all active compounds assessed in the initial experiments. Two main conclusions can be drawn. First, estradiol acts as an antagonist to vancomycin. Second, changes were more frequently observed in dual-species biofilms. It is also important to note the absence of correlation between the results of CV staining and CFU counts. This suggests that active compounds may affect the toughness biofilm, resulting in more or less biomass on the PTFE cubes after rinsing procedures.

### 3.4. Confocal Microscopy of L. paracasei and S. aureus Biofilms

Confocal imaging of biofilms grown in glass-bottom microtiter plates revealed that estradiol significantly inhibited both species (see Figure 4 and Appendix A for representative images). The inhibitory effect on *L. paracasei* is consistent with previous findings [33]. In monospecies biofilms, estradiol decreased the biomass density by 39% for *L. paracasei* and by 55% for *S. aureus* (Figure 4A). Ethanol slightly stimulated *L. paracasei*.

In dual-species biofilms, *L. paracasei* became even more sensitive to estradiol, with biomass density decreased by 19.7% (Figure 4B). Unlike *Micrococcus luteus* [33], *S. aureus* did not stimulate *Lactobacillus* biofilm formation or reverse estradiol’s effect. Instead, the presence of *S. aureus* enhanced inhibition. The combined effect of estradiol and vancomycin on *L. paracasei* was similar to the effect of estradiol. The combination of ethanol and vancomycin did not produce additional or antagonistic effects. Responses on glass-bottom plates mirrored those on PTFE cubes for *L. paracasei*.

For *S. aureus*, the combined effects were stronger on glass than on PTFE. Ethanol reduced vancomycin inhibition. Vancomycin lowered biomass density by 78% of the control value, whereas vancomycin plus ethanol increased by 20% relative to vancomycin (Figure 4A). Adding estradiol to vancomycin decreased biomass by 59.4% relative to the combination of vancomycin and ethanol. On PTFE cubes, the same combination had a stimulatory effect, indicating that estradiol action depends on the carrier surface.

Three points were evident in dual-species biofilms. First, *S. aureus* grew less than in monospecies biofilms, which is consistent with antagonism from *L. paracasei*. Its biomass density decreased by 63%; (Figure 4B). Second, *S. aureus* became less sensitive to vancomycin, with no inhibition was detected. Third, the stimulatory effect of vancomycin plus ethanol was stronger than in monospecies biofilms. This made the difference between the combination of ethanol and vancomycin and the combination of estradiol and vancomycin more pronounced. In ’vancomycin plus ethanol’ samples the biomass density increased *S. aureus* by 28% in comparison with vancomycin-treated samples. Meanwhile, estradiol plus vancomycin decreased by 44%; Figure 4B. Estradiol was less inhibitory to *S. aureus* in dual-species than in monospecies biofilms.

Overall, CLSM revealed that the effects of estradiol depend on the surface material and on interspecies interactions in dual-species communities, which can alter sensitivity to active compounds. This may potentially be the result of different interactions between hydrophobic and hydrophilic molecules and surfaces. Hydrophilic glass seems to be a worse surface for the physicochemical adhesion of organic molecules, especially hydrophobic ones like estradiol or different components of the cell envelope. Hydrophilic glass also seems to be a worse surface for the formation of a conditioning layer—a layer of organics that covers the surface and accommodates cell adhesion. Estradiol seems to be a cell surface-interacting molecule, and surface nature may be important. As we will show below, estradiol’s interaction with the surface is indeed important for its impact on cells.

As was shown above, estradiol and ethanol can modulate vancomycin efficacy against *L. paracasei* and *S. aureus*.

### 3.5. Monospecies and Dual-Species Biofilms in 96-Well Microtiter Plates

#### 3.5.1. The Effect of Estradiol Absorbed on the Surface of Wells

First, we studied how estradiol molecule adhesion to a carrier surface could influence bacterial growth. To accomplish this, we compared the effects of estradiol after standard inoculation under both aerobic and anaerobic conditions, as well as after pre-adhesion to a sterile medium overnight.

One notable finding was that planktonic cultures appeared to be more sensitive in this format (see Figure 5, Figure 6 and Figure 7). Several mechanisms could contribute to this phenomenon. First, some of the biomass may have been biofilm that detached during handling and was misread as planktonic optical density (OD). However, CV staining did not reveal proportional differences, suggesting that this is not the sole explanation. Second, compounds may interfere with the planktonic-to-biofilm transition rather than with proliferation. Third, changes in aggregation and matrix toughness could alter resistance to pipetting, resulting in different aggregate sizes and consequently, OD. The most likely explanation is a combination of these factors.

First, we analyzed S. aureus 209P (Figure 5). Planktonic cultures in 96-well plates were much more susceptible to all the active compounds after standard inoculation, in contrast to the system with PTFE cubes (Figure 5A). Ethanol strongly stimulated the growth of planktonic cultures under aerobic conditions. In the absence of oxygen, however, the effect was minimal. The antibiotic tended to stimulate the growth of planktonic *S. aureus*, especially under anaerobic conditions. Interestingly, estradiol significantly reduced ethanol-mediated stimulation under aerobic conditions, yet it stimulated planktonic growth under anaerobic conditions. A combination of vancomycin and ethanol under aerobic conditions negated the stimulatory effects of both compounds. In contrast, a combination of vancomycin and estradiol stimulated growth under aerobic conditions. Under anaerobic conditions, estradiol and vancomycin had a strongly inhibitory effect.

Interestingly, no effects of the active compounds were found in the case of pre-adhered estradiol and S. aureus planktonic growth. Vancomycin tended to stimulate growth; however, the variability of the data does not allow us to consider this difference statistically significant. Nevertheless, significant effects were only observed in biofilms after the pre-adhesion of estradiol (Figure 5B). Estradiol slightly decreased biofilm growth; when combined with vancomycin, inhibition was more pronounced.

Thus, estradiol and vancomycin act as antagonists in S. aureus, as do ethanol and estradiol. This is consistent with previous results [33].

*L. paracasei* was less sensitive to the active compounds under standard conditions (see Figure 6). Estradiol tended to stimulate planktonic growth in an anaerobic atmosphere. However, when combined with vancomycin, estradiol reduced growth (Figure 6A). The pronounced effects of the active compounds were only observed after microplates were pre-incubated overnight with RCM. The effects of the active compounds on planktonic and biofilm growth were similar for *L. paracasei.* The effects on biofilms were consistent with previous data: ethanol stimulated biofilms, while estradiol inhibited them (Figure 6B). Planktonic cultures were affected in the opposite way. The combination of vancomycin and ethanol stimulated planktonic growth, while estradiol reduced this effect (Figure 6A). However, biofilms were not susceptible to these combinations (Figure 6B).

Thus, estradiol and ethanol acted antagonistically with respect to both *L. paracasei* and *S. aureus*, as did vancomycin and estradiol. Preincubating the plates with RCM resulted in more pronounced effects of all the active compounds. This suggests that the organics in the medium create specific conditions on the well surface that modulate biofilm growth and susceptibility. Similar to *S. aureus*, *L. paracasei* biofilms were more sensitive to the hormone when it was pre-adsorbed on the surface.

Both dual-species planktonic cultures and biofilms (Figure 8) were more sensitive to the active compounds under standard conditions and after pre-incubation. Under an aerobic atmosphere, the dual-species communities resembled *S. aureus* mono-species planktonic cultures and biofilms. Under anaerobic conditions and after estradiol pre-adhesion, the behavior of the communities resembled a combination of the monospecies. However, estradiol was active while adhered to the well surface as well as when inoculated standardly. Additionally, the effects of estradiol and vancomycin, as well as ethanol and estradiol, were antagonistic in all cases.

#### 3.5.2. The Nature of Estradiol Solvent Is Important for Estradiol Effects After Pre-Adhesion

The next step was to understand how the effect of absorbed estradiol potentially depends on the liquid properties. Organics compounds dissolved in the RCM may form a conditioning layer that modulates the ability of cells to adhere to the surface. Additionally, this organic layer may alter the hormone’s ability to affect bacterial cells and adhere to the surface carrier.

We compared three types of solvents for estradiol. The first was RCM, followed by sterile PS and ethanol. Monospecies planktonic cultures of *S. aureus* were stimulated by vancomycin in the case of pre-incubation both in RCM and PS (Figure 8). However, the effect was more statistically reproducible in the case of PS. The organic material forming the conditioning layer is potentially less stable; therefore the resulting stimulation was not statistically significant despite the higher numeric difference. Regarding PS, several potential explanations for vancomycin-mediated stimulation can be proposed. First, in PS the residual organics can form a thinner, more strale layer that affects the bacterial growth. Second, there was slight dilution of the medium; however, it does not seen to be significant. Finally, it could be a combination of those two mechanisms, which should be the subject of future investigations.

Pre-incubation in ethanol led to the most pronounced effects of estradiol on monospecies *S. aureus* planktonic cultures, which can be explained by the higher efficiency of estradiol adhesion. Additionally, despite rinsing the wells, residual micro concentrations of ethanol may affect the growth and biofilm structure. Nevertheless, biofilms of *S. aureus* became less sensitive to estradiol and other compounds in case of PS and ethanol than in case of RCM (Figure 7). Therefore, the conditioning layer is an important factor for estradiol activity. While adhered to the surface, the hormone and other compounds affect the biofilm-planktonic transformation and biofilm structure density.

*L. paracasei* was more sensitive to pre-adhered estradiol both in planktonic and biofilm forms (Figure 9). In comparison with *S. aureus*, the nature of the solvent for estradiol was crucial. Furthermore, it was demonstrated that the potential conditioning layer and organics in RCM shifted the effect of the hormone. In RCM, the combination of estradiol and vancomycin inhibited the growth of planktonic cultures (Figure 9A) in comparison with the combination of vancomycin and ethanol. Meanwhile, biofilms were unaffected (Figure 9B). However, while pre-incubated in PS or ethanol, estradiol tended to stimulate biofilms, but a combination of estradiol and vancomycin inhibited them relatively to a combination of vancomycin and ethanol (Figure 9B). The effects on planktonic cultures were strictly opposite (Figure 9A), and the combination of vancomycin with estradiol had a strong stimulatory effect.

In dual-species communities (Figure 10), planktonic cultures were also more sensitive to active compounds. However, several special features were observed. First is that in the case of RCM and PS pre-incubation, the tendencies in the planktonic cultures and biofilms were closer to those in *L. paracasei* monospecies biofilms. In the case of ethanol pre-incubation, however the tendencies were closer to those in *S. aureus* (Figure 10A). Dual-species biofilms were stimulated by estradiol in the case of ethanol pre-incubation (Figure 10B).

Therefore, some conclusions can be drawn. First, estradiol was shown to affect bacteria and their biofilms when adsorbed onto a carrier surface. Second, the presence of organics in the liquid is important for the efficacy of estradiol and for susceptibility to other active compounds. Third, estradiol and vancomycin are antagonists, as are ethanol and estradiol. Fourth, the active compounds potentially affect the biofilm density and susceptibility to pipetting, and the shift between the biofilm and planktonic forms.

Figure 11 presents a general scheme of estradiol action under different conditions. It illustrates the higher efficacy of the active compounds under pre-adhesion conditions and the higher susceptibility of dual-species communities to all of them.

#### 3.5.3. The Comparison of *L. paracasei* and *L. gasseri* Susceptibility to Estradiol and Vancomycin

To analyze the effects of estradiol, vancomycin, and their combinations on lactobacilli, we studied vaginal *L. gasseri* ATCC 33323 and its community with *S. aureus* 209P. First, we tested the bacteria in an aerobic atmosphere (Figure 12). We observed that monospecies planktonic cultures of *L. gasseri* were much more sensitive to active compounds, especially ethanol (stimulation: 85%; Figure 12A). Estradiol decreased the planktonic OD, as did the combination of ethanol and vancomycin. Interestingly, vancomycin itself did not affect *L. gasseri*. Biofilms of *L. gasseri* were less affected (Figure 12B); however, they were more susceptible to the combination of estradiol and vancomycin than biofilms of *L. paracasei*.

Dual-species biofilms of *L. paracasei* and *S. aureus* were unaffected by the active compounds. However, biofilms of *L. gasseri* and *S. aureus* were inhibited by a combination of vancomycin and estradiol. Dual-species planktonic cultures were susceptible to ethanol stimulation, estradiol inhibition, and their combinations with vancomycin. As with monospecies *L. gasseri* planktonic cultures, the combination of estradiol and vancomycin had a stimulatory effect, especially in the *S. aureus* and *L. paracasei* community. Dual-species biofilms of *S. aureus* and *L. gasseri* reacted oppositely to the active compounds.

Under anaerobic conditions, the general situation was quite similar to that under aerobic conditions. However, *L. gasseri* monospecies biofilms were more susceptible to active compounds. Estradiol combined with vancomycin inhibited planktonic growth (Figure 13A) and stimulated biofilms (Figure 13B). Thus, the response of planktonic cultures and biofilms was opposite. Dual-species planktonic cultures and biofilms behaved similarly.

Consequently, the effects of estradiol, vancomycin, and their combination depend on the species of *Lactobacillus* present. Nevertheless, dual-species communities with *S. aureus* behaved similarly in the presence of both *L. paracasei* and *L. gasseri*, which suggests that staphylococci significantly affect the community. Additionally, it is important to note that estradiol increased the inhibitory activity of vancomycin in both communities.

### 3.6. Lipid Composition of Cells and Biofilm Matrix in Lactobacilli and Its Modulation by Estradiol

The biofilm matrix was extracted correctly, and the LDH test was negative. To investigate the biochemical changes induced by ethanol and estradiol, we analyzed the lipid profiles of two *L. paracasei* strains, AK508 (kvass-derived) and 27W (gut-derived), focusing on membrane lipids (Figure 14A,B), storage lipids (Figure 14C,D), and extracellular matrix lipids (Figure 14E,F). Numerical values are provided in Figure 6.

Overall, both strains responded similarly to estradiol. In membranes, ethanol increased phosphatidylcholines, whereas estradiol decreased them (Figure 14A,B). Estradiol increased membrane sterols, however, total membrane lipids were lower in the presence of the hormone, especially at 2.2 × 10^−8^ M.

Total storage lipids were higher in both strains with estradiol (Figure 14C,D), primarly due to increases in diacylglycerols and unidentified lipid fractions. Ethanol also had a stimulatory effect, notably on sterol esters in 2strain 7W (Figure 14D). Estradiol increased sterol esters in the extracellular matrix, in a strain- and dose-dependent manner and stimulated glycolipids in strain 27W (Figure 14F). Meanwhile, ethanol decreased glycolipids in AK508 (Figure 14E). Overall, estradiol increased total matrix lipids at 2.2 × 10^−10^ M and decreased them at 2.2 × 10^−8^ M. Thus, estradiol and ethanol had opposite effects on lipid synthesis pathways in lactobacilli. It is important to note that estradiol usually inhibited the growth of *L. paracasei* biofilms at a concentration of 2.2 × 10^−10^ M while increased lipid amounts. Therefore, *L. paracasei* biofilms potentially become more “saturated” with lipids in the presence of the hormone.

### 3.7. Antibacterial Activity of Monospecies and Dual-Species Biofilms in the Presence of Estradiol, Ethanol, and Vancomycin

We measured antibacterial activity using the agar block method. The activities varied by test organism and were occasionally absent in some repeats. Although the bacterial lawns appeared stable and identical in all samples, significant variations in activity were observed. This could result from various factors, including minimal shifts in the atmosphere’s composition (e.g., poorly sealed bags), minor alterations in the medium’s pH, or minor errors in the medium preparation process. It could also be a consequence of the strain’s unstable behavior.

Across six independent experiments, *L. paracasei* AK508 exhibited consistent activity against *Micrococcus luteus* C01, *M. luteus* HB, *Pseudomonas chlororaphis* 449, and *Kytococcus schroeteri* H01 (with no zero repeats; see Figure 15A,D), though estradiol, ethanol, and vancomycin did not significantly affect the diameter of the inhibition zones. Monospecies *L. paracasei* biofilms were active against *Pseudomonas aeruginosa* PAO1, in four repeats for control, ethanol, and vancomycin, and in five repeats for estradiol, vancomycin plus ethanol, and vancomycin plus estradiol. Estradiol increased the mean inhibition zone from 10 to 11 mm (Figure 15A). *Staphylococcus epidermidis* was generally insensitive, with activity detected in only one of six repeats (Figure 15D). *Escherichia coli* K-12 and *Candida* spp. were not inhibited (Figure 15D).

*S. aureus* monospecies biofilms exhibited weaker antibacterial activity. *M. luteus* HB was the most sensitive indicator strain, with inhibition detected in three out of six repeats, as well as the largest zone diameters (Figure 15B,E). *P. chlororaphis* was also sensitive. *M. luteus* C01 was less susceptible than HB. For *K. schroeteri*, at least five of the six repeats were negative, precluding conclusions.

Dual-species biofilms exhibited mixed activities and did not inhibit *S. epidermidis* (see Figure 15C,F). There were no significant changes in zone diameters were observed with active compounds (Figure 15C), though treatments generally reduced the proportion of zero repeats (Figure 15F).

Overall, antibacterial activity was unstable and largely independent of the compounds, contrasting with the stable activity previously reported for *L. paracasei* [33]. These differences may be due to the lower inoculum OD used in the present study or small deviations in the protocol between experiments.

### 3.8. Differential Gene Expression in S. aureus

In *S. aureus* monospecies biofilms, all tested compounds and combinations reduced expression of all resistance genes detected in *S. aureus* 209P. There was no clear differences among estradiol, ethanol, and vancomycin (Figure 16A). Ethanol and vancomycin generally exhibited antagonistic effects, and estradiol antagonized ethanol while demonstrating an additive effect with vancomycin. There was high variability, and most differences were not statistically significant (*p* > 0.05).

In contrast, in dual-species biofilms with *L. paracasei*, the treatment effects switched. Ethanol remained being a repressor. Estradiol antagonized ethanol, restoring expression to control levels for *norC* and *vanTG*. Vancomycin shifted expression from downregulation in monospecies biofilms to slight upregulation in dual-species biofilms for most genes except *mepR*. The combination of ethanol and vancomycin strongly enhanced the effects of vancomycin (Figure 16B). Estradiol similarly enhanced the expression of the efflux pump genes *lmrS* and *sdrM*.

Two conclusions follow. First, the presence of *L. paracasei* is crucial for *S. aureus* transcriptional responses. Compounds that downregulate resistance genes in monospecies biofilms often upregulate them in dual-species communities. Second, as in other phenotypic assays, ethanol, estradiol, and vancomycin commonly acted antagonistically. The combination of vancomycin and ethanol can upregulate genes associated with resistance to multiple classes of antibiotics, including glycopeptides, tetracyclines, cephalosporins, fluoroquinolones, macrolides, phenicols, and aminoglycosides. Most of these targets encode efflux pumps (except *vanTG*), which is consistent with the effects of these compounds on the cell envelope.

Only two resistance genes were found in *L. paracasei* AK508. The first is *vanTG*, and the second is *qacJ*, an efflux pump that removes quaternary ammonium compounds. Since their differential expression was unstable in all cases, we decided to focus on *S. aureus*. However, the presence of *vanTG* in *L. paracasei* is one reason for its high vancomycin resistance.

## 4. Discussion

The ability of hormones to modulate biofilm growth is now well recognized. Accordingly, the central question has shifted from whether hormones act on bacteria to how they do so and how their effects can be harnessed. Another practical approach is to explore the use of host hormones to regulate microbial communities and enhancer or modulate conventional antibiotics. This is particularly relevant for hormones, such as steroids, which are widely used as pharmacological substances. The present study examined the estradiol’s potential to modulate vancomycin susceptibility in selected Gram-positive bacteria.

Consistent with previous observations [33], estradiol and ethanol acted as antagonists at the tested concentrations. Estradiol has been reported to modify *Pseudomonas aeruginosa* membranes, potentially through the *mucABCD* operon, thereby shifting membrane lipid polarity and fluidity [19]. However, protein BLAST (version BLAST+ 2.17.0, 21 July 2025) queries [52] did not detect mucB-like proteins in the lactobacilli or staphylococci examined here. Nonetheless, the observed estradiol-associated shifts in *Lactobacillus* lipid composition suggest that a related membrane-centered mechanism may operate in Gram-positive bacteria. One plausible model is that estradiol and perhaps other steroids function as agonists for bacterial receptors that are not sequence homologs of classical eukaryotic steroid receptors or the Muc proteins of pseudomonads, but share similar three-dimensional features. Additionally, Clabaut et al. reported a protein that may act as a steroid sensor in lactobacilli. Clabaut et al. proposed a membrane lipid raft-associated SPFH domain-containing protein that shows homology with the eukaryotic estrogen-related receptor gamma (ERR3) and the estradiol-binding protein prohibitin-2 (PHB2) in *L. crispatus* [18]. Some *Mycobacterium* species have been described as having steroid-binding cytochromes P450 [53]. Steroid receptors are typically dimers with DNA-binding domains containing zinc finger motifs [54]. DNA-binding proteins with a similar 3D structure may be steroid- and estradiol-binding receptors in bacteria, including different LuxR solo receptors [55]. Nevertheless, this hypothesis warrants evaluation using molecular docking, high-resolution lipidomics, genetics, and mass spectrometry.

Estradiol regulated the behavior of *Staphylococcus aureus* and *Lactobacillus paracasei*, with more pronounced effects observed in dual-species biofilms than in monospecies biofilms. Under standard addition, *L. paracasei* monospecies biofilms on hydrophobic carriers such as PTFE and polystyrene were insensitive to estradiol under anaerobic conditions but were inhibited on glass, consistent with earlier work [33]. In dual-species biofilms, both species became more susceptible to estradiol, aligning with prior reports of interspecies modulation [38,56]. Notably, no switch from inhibition to stimulation of biomass was detected in the mixed communities studied here; estradiol’s remained consistent. This may reflect properties specific to the *S. aureus*–*L. paracasei* community. *S. aureus* may have a limited capacity to promote partner growth. Compared with *Micrococcus luteus*, *S. aureus* is taxonomically and physiologically closer to *L. paracasei*, which could intensify competition rather than facilitation.

An important finding is the antagonism between estradiol and ethanol, consistent with previous reports [33]. This antagonism may result from impairments in estradiol-mediated reaction pathways caused by ethanol in bacteria. In rats, such impairments were demonstrated at an ethanol concentration of approximately 1.2% [57]. Although our concentration was lower (0.06%), ethanol still has a significant impact on bacteria, even at such a low concentration [58]. This finding also supports the hypothesis of a steroid hormone receptor in *S. aureus* and *Lactobacillus*. Both ethanol and estradiol were antagonistic toward vancomycin. This antagonism is unlikely to result from direct chemical inactivation due to the non-stoichiometric molar ratios. Ethanol was present at 1.7 × 10^−3^ M, while vancomycin and estradiol were at 6.89 × 10^−7^ M and 2.2 × 10^−10^ M, respectively. Clinically, vancomycin is generally regarded as unaffected by alcohol consumption [56], and experimental work in *E. coli* and *S. aureus* has supported this view [59], although a historical case report described treatment failure in an elderly alcoholic patient [60]. These studies addressed therapeutic dosing. In contrast, the present study used a subinhibitory concentration of vancomycin (0.001 µg/mL) [61,62], which can stimulate biofilms. Under these conditions, ethanol increased the stimulatory action of vancomycin or diminished its inhibition. Given the clinical relevance of subinhibitory antibiotic exposure and the nontrivial incidence of alcohol use during treatment, these findings merit further investigation. qPCR results support this interpretation. Increased *vanTG* expression in *S. aureus* (a 29-fold increase, *p* < 0.05) may contribute to enhanced vancomycin tolerance in dual-species biofilms. This coincides with increased biomass in the simultaneous presence of ethanol and vancomycin. In *S. aureus* monospecies biofilms, attenuation of vancomycin-mediated inhibition was also observed in some systems (e.g., on glass), while the vancomycin plus ethanol combination showed no effect in others, which is consistent with the transcriptional data. Estradiol, acting as an antagonist of ethanol, typically countered the effect of ethanol plus vancomycin by switching stimulation to inhibition or the reverse. Together, these observations suggest a transcriptional basis for the regulatory effects of estradiol and ethanol.

Another key observation is that estradiol can adsorb to surfaces and regulate planktonic-to-biofilm switching and biofilm structure. In these experiments, estradiol altered the behavior of both species more strongly when pre-adsorbed than when added by the standard method. This suggests an interaction with cell surface structures during the initial attachment. Importantly, planktonic cultures in microtiter plates were much more sensitive to active compounds than cultures in other experiments, possibly due to methodological weaknesses. During sampling, standard pipetting can detach variable amounts of loosely adherent biomass. If compounds alter matrix density or aggregation, differences in OD_540_ may reflect changes in aggregate size and detachment rather than growth alone. Hence, important methodological considerations for future work avoiding the complete removal of the planktonic culture volume by pipetting and creating a diluted suspensions in a new microplate.

The solvent used for estradiol pre-adsorption affected the behavior of the bacterial community. Pre-adsorption from ethanol or RCM caused the dual-species community to resemble *S. aureus* monospecies biofilms. In contrast, prea-dsorption from physiological saline favored a *Lactobacillus*-like response. Additionally, in plates pre-incubated with physiological saline, *L. paracasei* biofilms were significantly stimulated by vancomycin despite the absence of exogenous organic compounds during pre-incubation. One possible explanation is the adsorption of volatile organic compounds from other cultures in the shared incubator into the saline, which were then adsorbed to the well surface. Ethanol would be an ineffective solvent for some of these volatiles, whereas RCM is rich in organics and may buffer their effects. This hypothesis requires dedicated testing, but highlights another methodological consideration: incubating laboratory plastics with small volumes of water-based liquids in proximity to actively growing cultures may allow volatile organics to dissolve and influence bacterial behavior.

Antibacterial assays against indicator organisms revealed considerable instability. This is unusual for lactobacilli, given prior reports of consistent activity [33]. The reduced stability may be due to the lower culture OD used in this study (1 instead of 2 used previously [33]) or to minor deviations from the standard protocol, such as differences in the tightness of anaerobic bags or in the uniformity of biomass spreading. Additionally, potential changes in the composition of the extracellular matrix (particularly its lipid compartment, as revealed in this study) may explain the instability of antibacterial activity. However, the biomass taken from the Petri dishes in matrix experiments appeared stable. Nevertheless, the most susceptible indicators, including micrococci, kytococci, and *Pseudomonas chlororaphis*, were inhibited by *L. paracasei* in all six repeats. Previously, it was suggested that the biofilm life form allowed *Lactobacillus* to exhibit more pronounced antimicrobial activity. However, it has been reported that different *Lactobacillus* species exhibit varying degrees of dependence on their antimicrobial activity when switching from a planktonic to a biofilm state [63]. Thus, *L. paracasei* may also exhibit such agile behavior.

Notably, estradiol and vancomycin tended to make *L. paracasei* activity more consistently detectable, increasing the number of positive outcomes, even though inhibition zone diameters changed little and combinations often reduced stability. The presence of *S. aureus* in dual-species communities reduced antibacterial activity, providing additional evidence for antagonism between community members and the modulatory roles of estradiol and vancomycin.

In summary, estradiol and ethanol often blocked the the effectiveness vancomycin in both monospecies and dual-species biofilms of *L. paracasei* and *S. aureus*. This was evident at the biomass and transcriptional levels. Estradiol’s effects on lactobacilli were dependent on the species and conditions, but the antagonism between vancomycin and estradiol was consistent across the three species examined. The stimulation of *S. aureus* biomass in dual-species biofilms by the combination of vancomycin and ethanol may be mediated, at least in part, by the upregulation of *vanTG* gene, an effect that is partially neutralized by estradiol. Estradiol was more effective when adsorbed to the biofilm carrier surface, acting as a regulator of lipid synthesis in lactobacilli. These findings underscore estradiol’s capacity to influence multispecies biofilms and support further investigation into its molecular targets and potential applications in microbiology and medicine.

## 5. Conclusions

Estradiol can regulate the growth and metabolism of *Staphylococcus aureus* and *Lactobacillus paracasei* biofilms with detectable effects at the transcriptional, metabolic, and population levels. The data are consistent with a mechanism involving interaction with surface-associated receptors or sensors. However, the specific molecular targets remain to be identified. Future work should define the receptor(s) for estradiol and delineate the pathways that mediate its multitarget actions.

Across the tested conditions, estradiol and ethanol acted as universal antagonists of vancomycin, altering the antibiotic response in both monospecies and dual-species biofilms. Therefore, estradiol represents a potential regulatory factor for bacterial communities and their susceptibility to antibiotics, with clear implications for microbiology and pharmaceutics, as well as a rationale for further mechanistic and applied investigations.

## Figures and Tables

**Figure 1 microorganisms-13-02777-f001:**
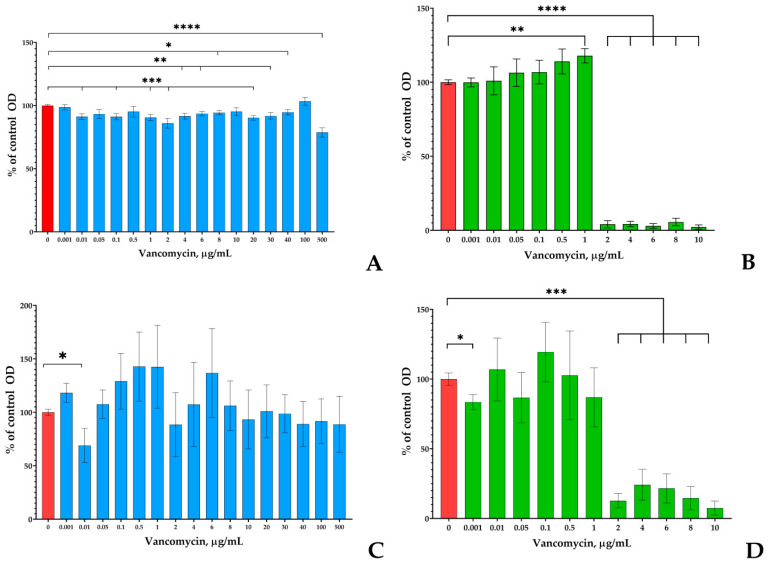
The effect of vancomycin at different concentrations on monospecies planktonic cultures (**A**,**B**) and biofilms (**C**,**D**) of *L. paracasei* AK508 (**A**,**C**) and *S. aureus* 209P (**B**,**D**) is shown. Biofilms were grown on PTFE cubes and stained with CV. * *p* < 0.05; ** *p* < 0.01; *** *p* < 0.005; **** *p* < 0.001.

**Figure 2 microorganisms-13-02777-f002:**
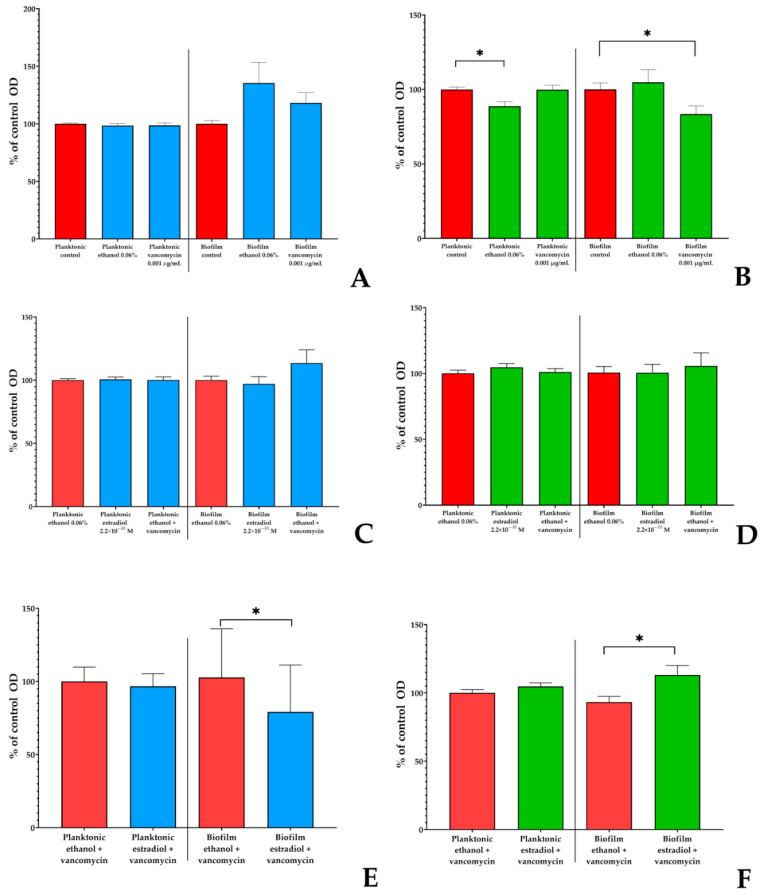
The effects of active compounds in monospecies biofilms of *L. paracasei* AK508 (**A**,**C**,**E**) and S. aureus 209P (**B**,**D**,**F**) grown on PTFE cubes and stained with CV. (**A**,**B**)—the effects of ethanol and vancomycin in comparison with controls without additions; (**C**,**D**)—the effects of estradiol and the combination of vancomycin and ethanol in comparison with ethanol-treated samples; (**E**,**F**)—the effect of the combination of vancomycin and estradiol in comparison with samples treated with the combination of ethanol and vancomycin * *p* < 0.05.

**Figure 3 microorganisms-13-02777-f003:**
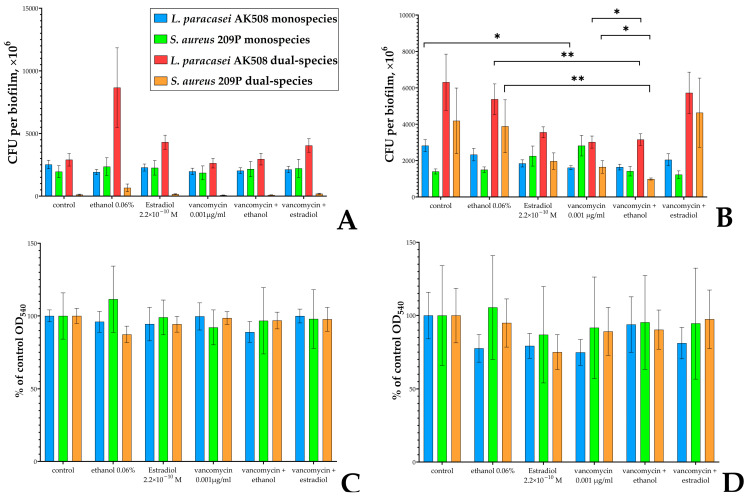
Effects of estradiol and vancomycin on CFU counts (**A**,**B**) and OD_540_ of formazan extracts (**C**,**D**) for *S. aureus* 209P and *L. paracasei* AK508. * *p* < 0.05; ** *p* < 0.01.

**Figure 4 microorganisms-13-02777-f004:**
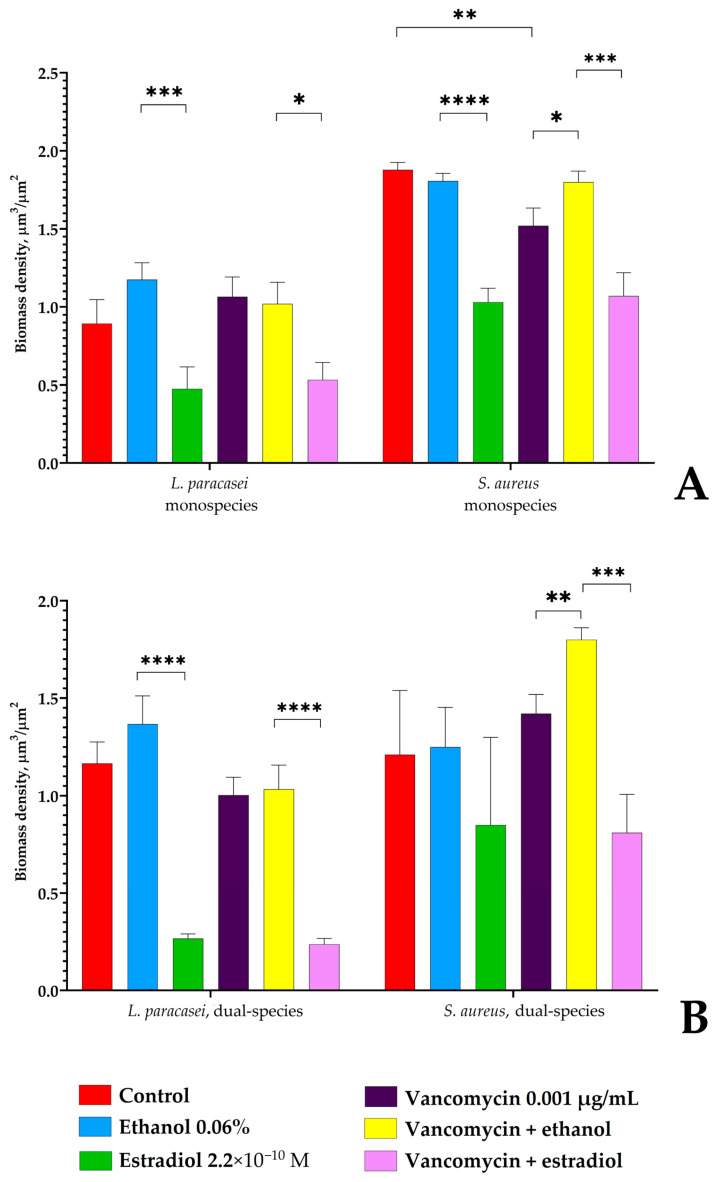
The effects of estradiol and vancomycin on average biomass density in monospecies (**A**) and dual-species (**B**) biofilms of *S. aureus* 209P and *L. paracasei* AK508 measured by CLSM. * *p* < 0.05; ** *p* < 0.01; *** *p* < 0.005; **** *p* < 0.001.

**Figure 5 microorganisms-13-02777-f005:**
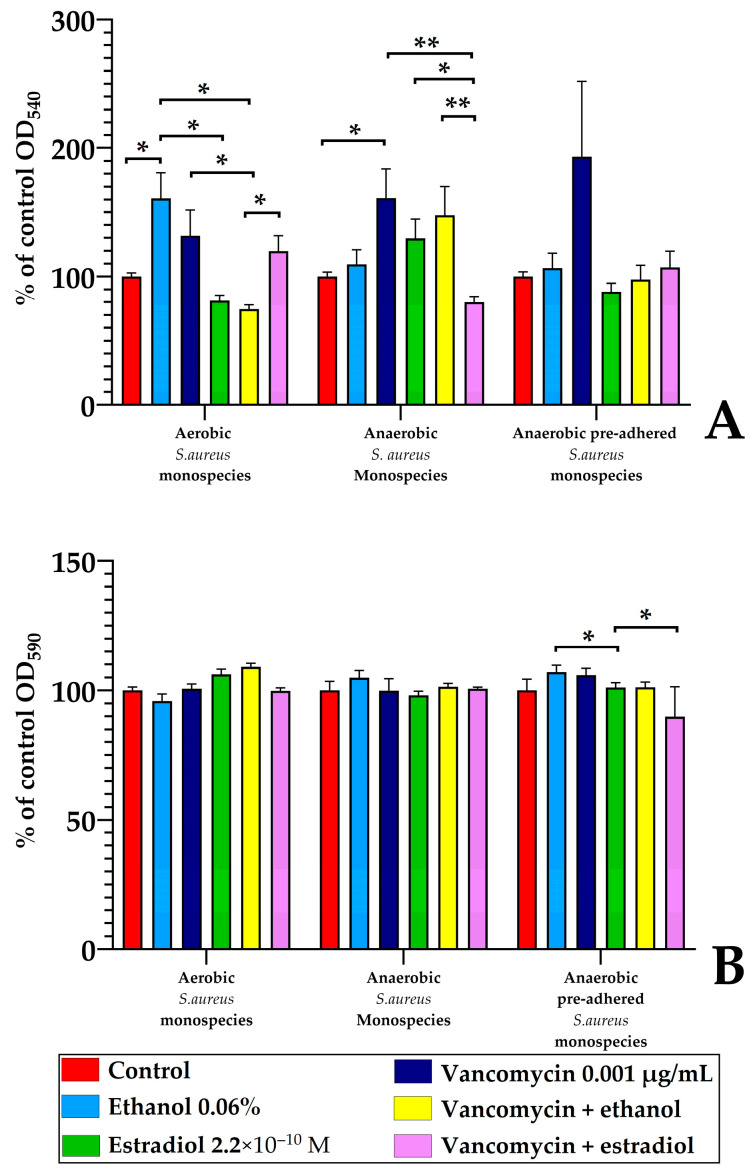
The effect of estradiol on monospecies *S. aureus* planktonic cultures (**A**) and biofilms (**B**) in 96-well microtiter plates. The incubation and pre-adhesion of estradiol were in the RCM medium. * means *p* < 0.05; ** means *p* < 0.01.

**Figure 6 microorganisms-13-02777-f006:**
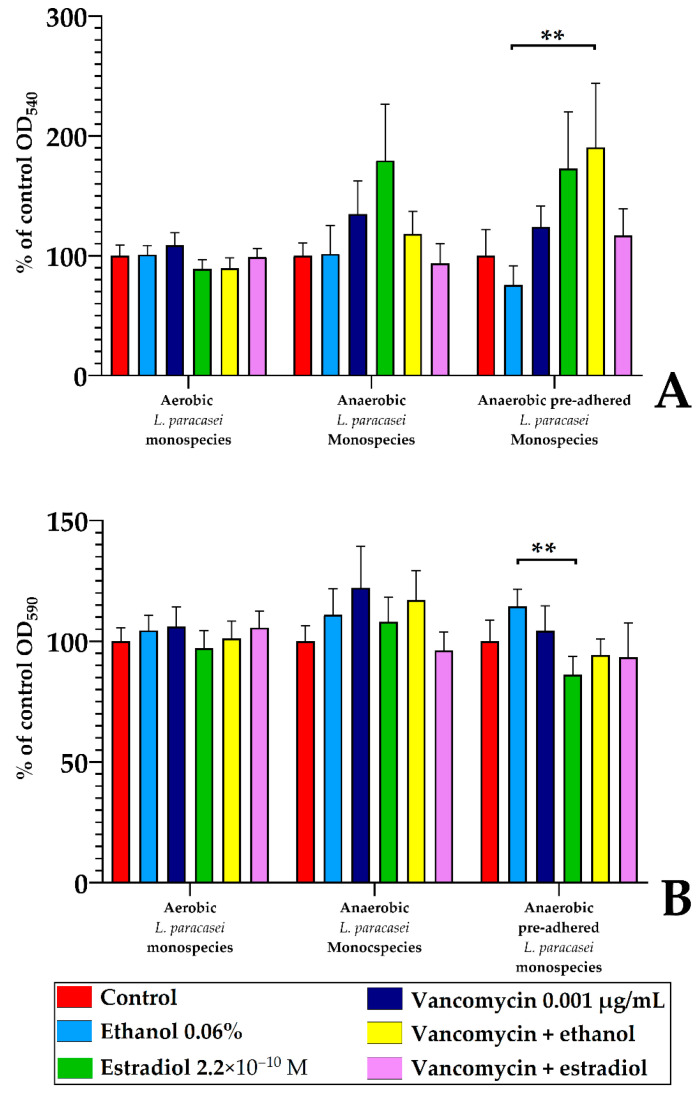
The effect of estradiol on monospecies *L. paracaasei* planktonic cultures (**A**) and biofilms (**B**) in 96-well microtiter plates. The incubation and pre-adhesion of estradiol were in the RCM medium. The incubation was in the RCM medium. ** means *p* < 0.01.

**Figure 7 microorganisms-13-02777-f007:**
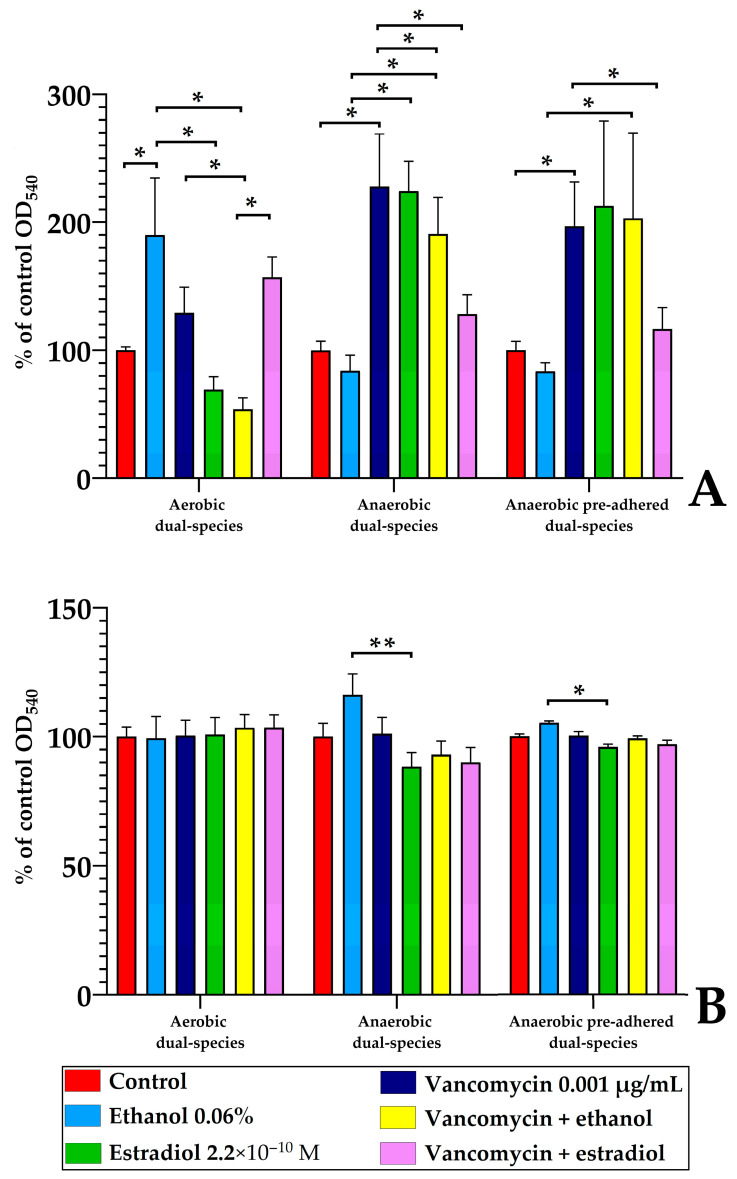
The effect of estradiol on dual-species planktonic cultures (**A**) and biofilms (**B**) in 96-well microtiter plates. The incubation and pre-adhesion of estradiol were in the RCM medium. The incubation was in the RCM medium. * means *p* < 0.05; ** means *p* < 0.01.

**Figure 8 microorganisms-13-02777-f008:**
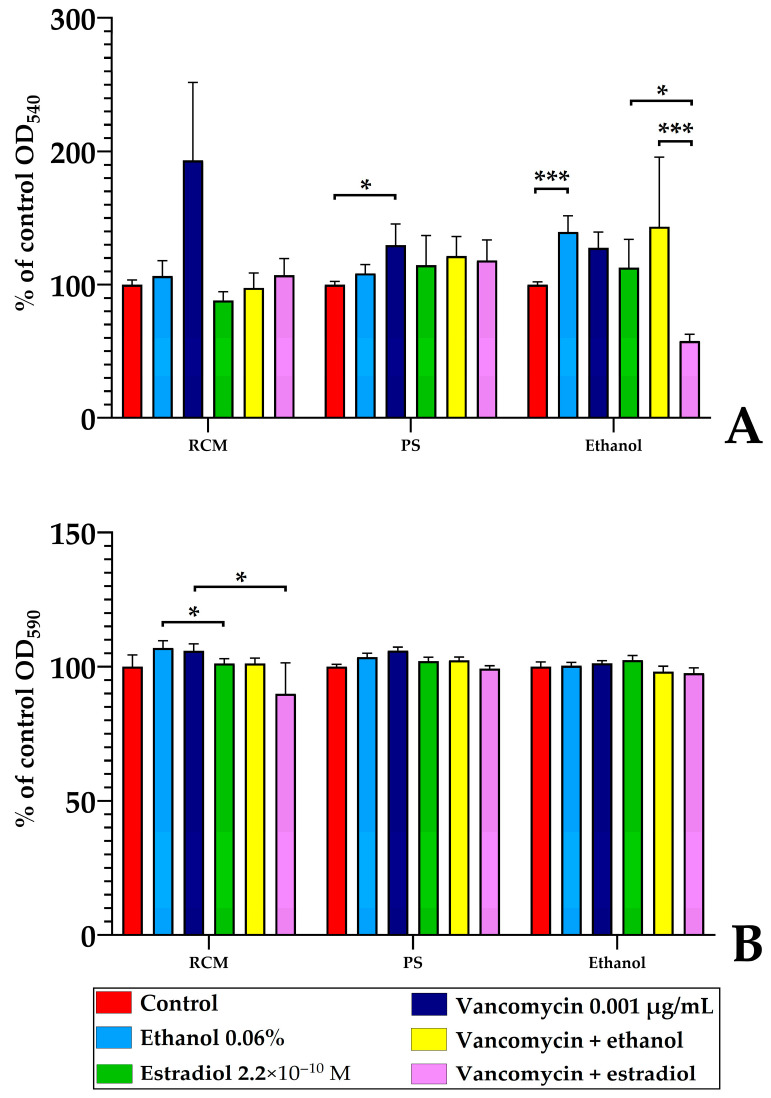
The impact of a solvent type on the effect of estradiol on *S. aureus* planktonic cultures (**A**) and biofilms (**B**) after overnight pre-adhesion. PS—physiological saline. * means *p* < 0.05; *** means *p* < 0.005.

**Figure 9 microorganisms-13-02777-f009:**
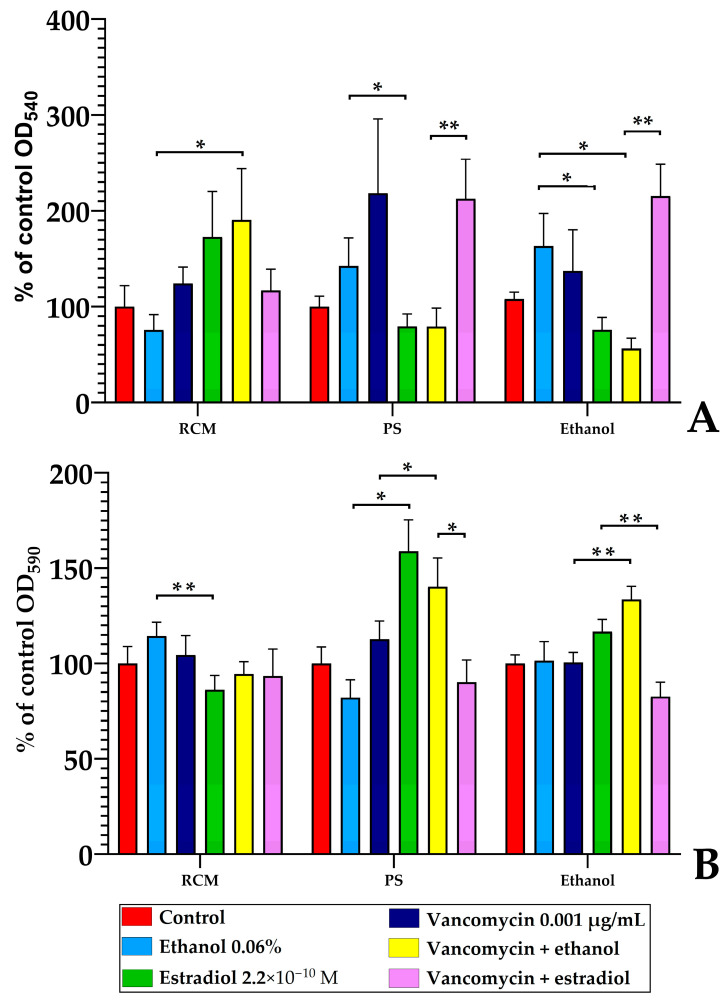
The impact of a solvent type on the effect of estradiol on *L. paracasei* planktonic cultures (**A**) and biofilms (**B**) after overnight pre-adhesion. PS—physiological saline. * means *p* < 0.05; ** means *p* < 0.005.

**Figure 10 microorganisms-13-02777-f010:**
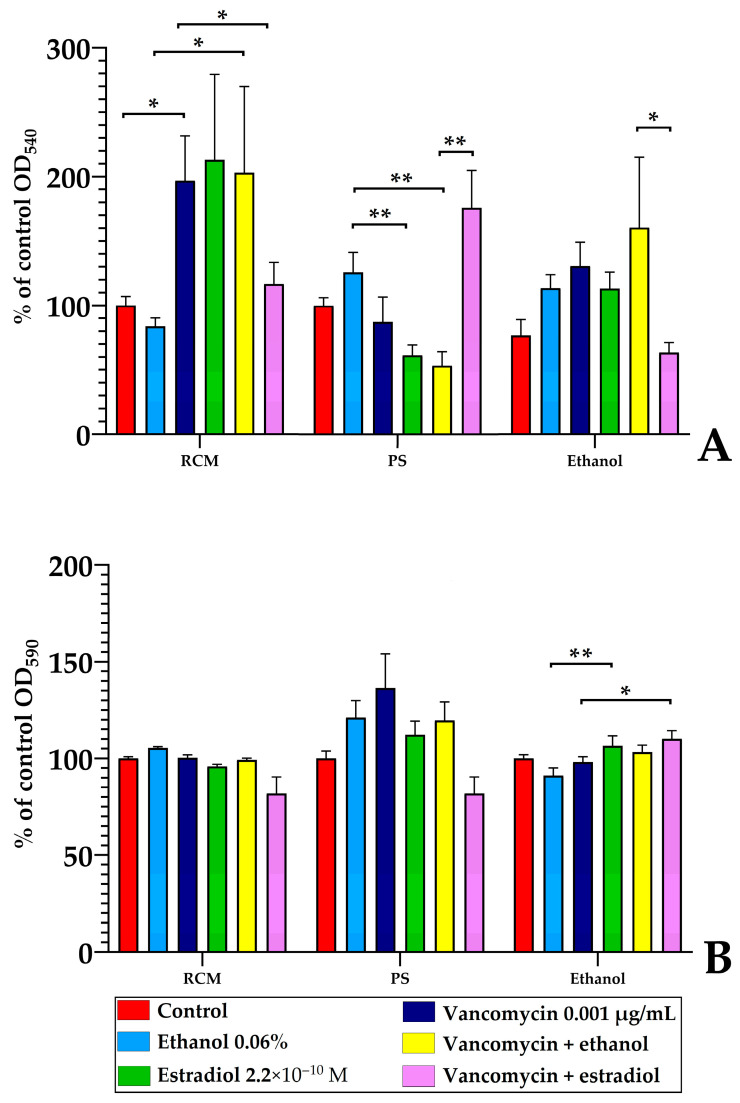
The impact of a solvent type on the effect of estradiol on dual-species planktonic cultures (**A**) and biofilms (**B**) after overnight pre-adhesion. PS—physiological saline. * means *p* < 0.05; ** means *p* < 0.01.

**Figure 11 microorganisms-13-02777-f011:**
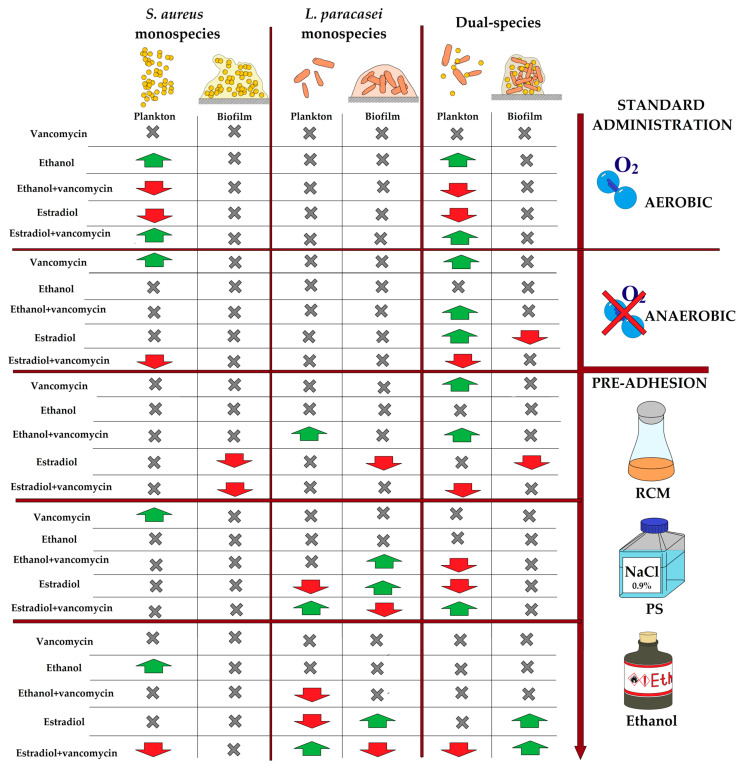
A general scheme showing the effects of estradiol on monospecies and dual-species planktonic cultures and biofilms of *S. aureus* 209P and *L. paracasei* AK508 under different conditions. Green arrows indicate stimulation, red arrows indicate inhibition, and a gray cross indicates no significant effect.

**Figure 12 microorganisms-13-02777-f012:**
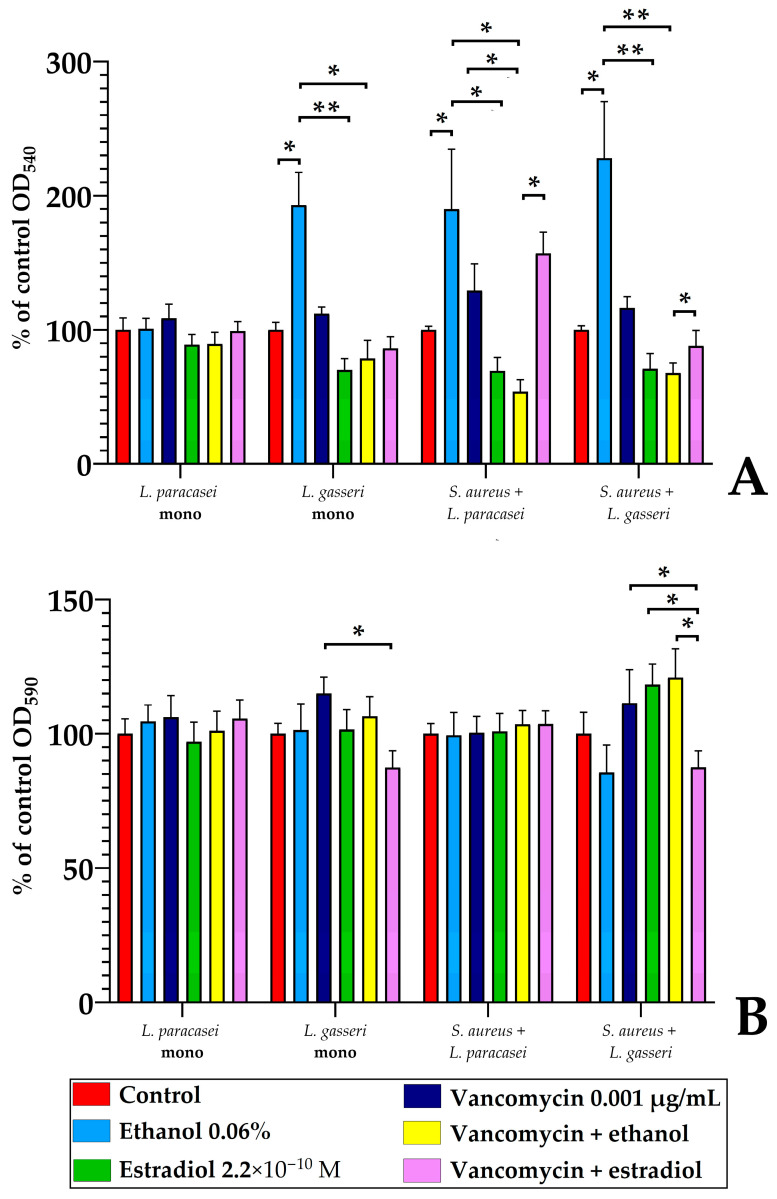
The comparison of monospecies planktonic cultures (**A**) and biofilms (**B**) of *L. paracasei*, and *L. gasseri* and their dual-species communities with *S. aureus* 209P. The cultivation was conducted under aerobic conditions. * means *p* < 0.05; ** means *p* < 0.01.

**Figure 13 microorganisms-13-02777-f013:**
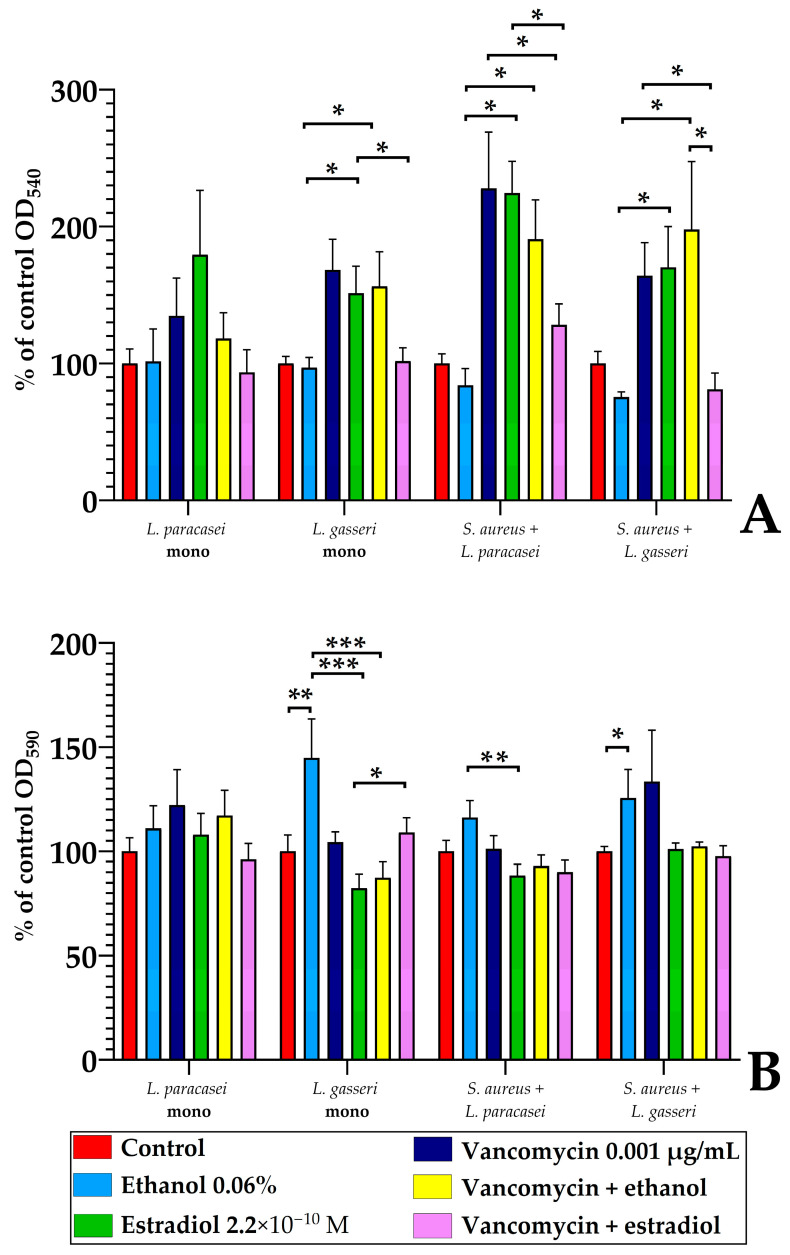
The comparison of monospecies planktonic cultures (**A**) and biofilms (**B**) of *L. paracasei*,and *L. gasseri* and their dual-species communities with *S. aureus* 209P. The cultivation was conducted under anaerobic conditions. * means *p* < 0.05; ** means *p* < 0.01; *** means *p* < 0.005.

**Figure 14 microorganisms-13-02777-f014:**
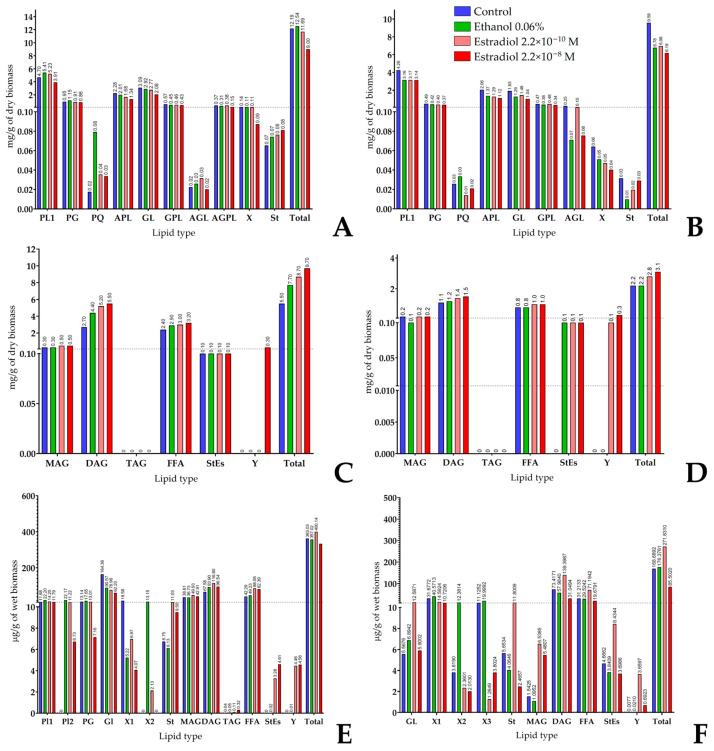
Changes in lipid composition of *L. paracasei* in the presence of estradiol. (**A**,**C**,**E**): AK508; (**B**,**D**,**F**): 27W. (**A**,**B**): membrane lipids; (**C**,**D**): storage lipids; (**E**,**F**): extracellular matrix lipids. Pl, phospholipids; PG, phosphatidylglycerols; PC, phosphatidylcholines; APL, aminophospholipids; GL, glycolipids; GPL, glycophospholipids; AGL, aminoglycolipids; AGPL, aminoglycophospholipids; St, sterols; DAG, diacylglycerols; FFA, free fatty acids; MAG, monoacylglycerols; StEs, sterol esters; TAG, triacylglycerols; X, Y, unidentified lipids. The dash lines indicates scale gaps on the Y axis.

**Figure 15 microorganisms-13-02777-f015:**
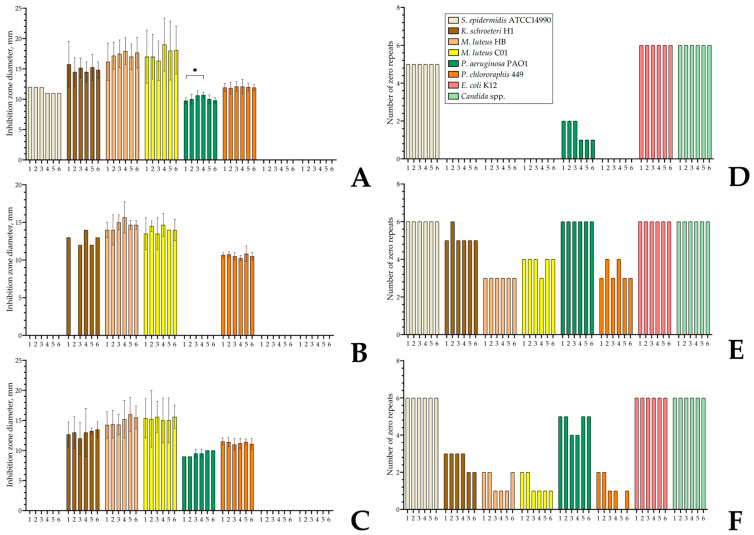
Antibacterial effects of monospecies *L. paracasei* (**A**,**D**), *S. aureus* (**B**,**E**), and their dual-species biofilms (**C**,**F**). (**A**–**C**): inhibition zone diameters. (**D**–**F**): number of zero repeats. Conditions: 1 control, 2 ethanol, 3 vancomycin, 4 estradiol, 5 vancomycin plus ethanol, 6 vancomycin plus estradiol. * *p* < 0.05.

**Figure 16 microorganisms-13-02777-f016:**
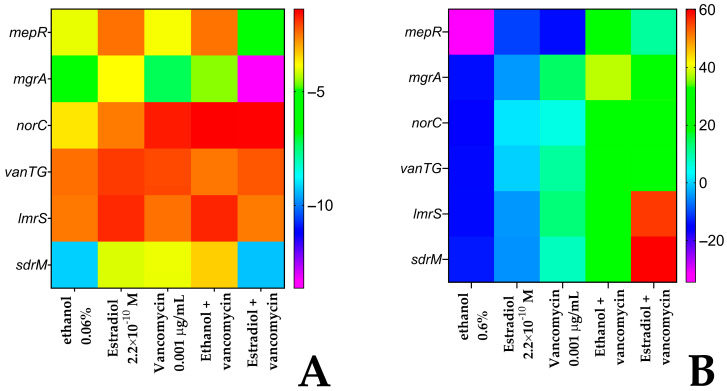
Differential expression of *S. aureus* resistance genes in the presence of active compounds. The heatmap shows how gene expression levels change compared to the control group. (**A**), monospecies biofilms. (**B**), dual-species biofilms.

**Table 1 microorganisms-13-02777-t001:** FISH probes used in this study.

Microorganism	Sequence 5′-3′	Fluorophore	Wavelength, nm	Reference
*S. aureus* 209P	GCCCCAAGATTACACTTCCG	FAM	488	[39]
*L. paracasei* AK508	GTATTAGCAYCTGTTTCCA	R6G	561	[40]

**Table 2 microorganisms-13-02777-t002:** Summary of the results of CFU counting, MTT and CV staining of *S. aureus* and *L. paracasei* monospecies and dual-species biofilms. “N/E means” no effect; “N/A” means not applicable; red color and “In” means inhibition; green color and “St” means stimulation; light-orange color and “TIn” means tendency to inhibition; light-green color and “TSt” means tendency to stimulation. Light-colored cells or colorless cells mean no statistical significance of differences or no difference.

	OD_540_ of an Inoculum	Analysis	Ethanol	Estradiol	Vancomycin 0.001 µg/mL		
					Alone	+Ethanol	+Estradiol
*S. aureus* in monospecies biofilms	0.5	CFU	N/E	N/E	N/E	N/E	N/E
		MTT	N/E	N/E	N/E	N/E	N/E
		CV on PTFE	N/E	N/E	In	N/E	St
	2	CFU	N/E	TSt	TSt	TIn	TIn
		MTT	N/E	N/E	N/E	N/E	N/E
*S. aureus* in dual-species biofilms	0.5	CFU	TSt	TIn	TIn	TIn	TSt
		MTT	N/A	N/A	N/A	N/A	N/A
	2	CFU	N/E	TI	TI	In	St
		MTT	N/A	N/A	N/A	N/A	N/A
*L. paracasei* in monospecies biofilms	0.5 (*S. aureus* 0.5)	CFU	TIn	TSt	N/E	N/E	N/E
		MTT	N/E	N/E	N/E	TIn	TSt
		CV on PTFE	TSt	N/E	TSt	TSt	In
	0.5 (*S. aureus* 2)	CFU	TIn	TIn	TIn	N/E	TSt
		MTT	TIn	N/E	TIn	TSt	TIn
*L. paracasei* in dual-species biofilms	0.5 (*S. aureus* 0.5) 0.5 (*S. aureus* 2)	CFU	TSt	TIn	N/E	TSt	TSt
		MTT	N/A	N/A	N/A	N/A	N/A
		CFU	TIn	TIn	TIn	In	St
		MTT	N/A	N/A	N/A	N/A	N/A

## Data Availability

The original contributions presented in this study are included in the article/Appendix A. Further inquiries can be directed to the corresponding authors.

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
