# Peer review of "Estradiol Modulates the Sensitivity to Vancomycin of *Lactobacillus paracasei* and *Staphylococcus aureus* Biofilms—Constituents of Human Skin and Vaginal Microbiota"

_microorganisms, 2025, doi:10.3390/microorganisms13122777_

Round 1

Reviewer 1 Report

Comments and Suggestions for Authors

The paper is interesting and can be accepted after revision.

  1. Authors mention that the antibacterial activity of L. paracasei was not stable between experimental repeats. Authors should clarify the possible reasons of this phenomenon. Is the RCM medium actually an appropriate medium for lactobacilli?
  2. Authors should discuss better possible effects of ethanol on bacteria. Why ethanol actually acts as an antagonist for estradiol? Also, is should be proved better the adequacy of ethanol concentration used. Ethanol is a strong effector, and may estradiol be dissolved in aqueous medium instead of ethanol?
  3. Authors should clarify is actually a concentration of estradiol 2.2×10-10 M an adequate for such an investigation? Actually, if we speak about different hormonal drugs, the concentration of estradiol is normally increased. Also, during the cycle it changes in women. Finally, in the tissues of female urogenital system it should be lower than in blood plasma. So, why 2.2×10-10 M?
  4. The text should be checked for grammar and typos.
  5. Figures should be redacted to make them easier for comprehension.

Author Response

Comments 1: Authors mention that the antibacterial activity of L. paracasei was not stable between experimental repeats. Authors should clarify the possible reasons of this phenomenon. Is the RCM medium actually an appropriate medium for lactobacilli?

Response 1: Thank you for the remark! We added some thoughts in the text (lines 1131-1135)

“Although the bacterial lawns appeared stable and identical in all samples, significant variations and instability in activity were observed. This could be a result of various fac-tors, including minimal shifts in the atmosphere's composition (e.g., poorly sealed bags), minor alterations in the medium's pH, or minimal errors in the medium preparation pro-cess. It could also be a consequence of unstable behavior of the strain itself.”

Also, we added the text in lines 1317-1326

“Additionally, potential changes in the composition of the extracellular matrix (particular-ly its lipid compartment, as revealed in this study) may explain the instability of antibac-terial activity. However, the biomass taken from the Petri dishes in matrix experiments appeared stable. Nevertheless, the most susceptible indicators, including micrococci, kytococci, and Pseudomonas chlororaphis, were inhibited by L. paracasei in all six repeats. Pre-viously, it was suggested that the biofilm life form allowed Lactobacillus to exhibit more pronounced antimicrobial activity. However, it has been reported that different Lactobacil-lus species exhibit varying degrees of dependence on their antimicrobial activity when switching from a planktonic to a biofilm state [62]. Thus, L. paracasei may also exhibit such agile behavior.”

Comments 2: Authors should discuss better possible effects of ethanol on bacteria. Why ethanol actually acts as an antagonist for estradiol? Also, is should be proved better the adequacy of ethanol concentration used. Ethanol is a strong effector, and may estradiol be dissolved in aqueous medium instead of ethanol?

Response 2: Thank you! We added a paragraph in the discussion (lines 1251-1257)

An important finding is the antagonism between estradiol and ethanol, consistent with previous reports [33]. This antagonism may result from impairments in estradiol-mediated reaction pathways caused by ethanol in bacteria. In rats, such impairments were demonstrated at an ethanol concentration of approximately 1.2% [57]. Although our concentration was lower (0.06%), ethanol still has a significant impact on bacteria, even at such a low concentration [58]. This finding also supports the hypothesis of a steroid hormone receptor in S. aureus and Lactobacillus.

Comments 3: Authors should clarify is actually a concentration of estradiol 2.2×10-10 M an adequate for such an investigation? Actually, if we speak about different hormonal drugs, the concentration of estradiol is normally increased. Also, during the cycle it changes in women. Finally, in the tissues of female urogenital system it should be lower than in blood plasma. So, why 2.2×10-10 M?

Response 3: Thank you! We added a paragrath in the text (lines 156-159)

“Blood plasma concentration is a more stable parameter because the concentration of es-tradiol in vaginal mucus is highly variable. It depends not only on the phase of the men-strual cycle, but also on vaginal health and the use of hormonal drugs”

Comments 4: Figures should be redacted to make them easier for comprehension.

Response 4: Thank you! We redacted some figures in the text especially .in the subsection of microplate cultivation

4. Response to Comments on the Quality of English Language

Point 1: The text should be checked for grammar and typos.

Response 1: Thank you! We addressed the remark

Reviewer 2 Report

Comments and Suggestions for Authors

The manuscript is dedicated to an actual problem – modification of sensitivity to antibiotics in biofilms by different compounds. Authors investigated rather unusual object – binary biofilms of Lactobacillus paracasei and Staphylococcus aureus and their sensitivity to estradiol and vancomycin. Authors used a number of methods to study biofilms – standard CV staining, CFU counting, CLSM, some biochemical experiments (study of lipids) and some molecular approaches (PCR). Hence, biofilms seem to be studied comprehensively.

However, some points should be elucidated better.

  • The authors should better discuss how relevant their data actually is. Is vancomycin an antibiotic really widely used in vaginitis treatment? How actually frequent are staphylococcal infections of female urogenital tract?
  • Why authors actually used Lactobacillus paracasei? This is not a dominant part of vaginal microbiota. L. gasseri seems to be much more relevant object, however, it was not studied as precisely as L. paracasei was.
  • In the section of differential gene expression authors studied S. aureus genes. However, what about L. paracasei? It should be also checked better or at least mentioned in the discussion
Comments on the Quality of English Language

The English style should be checked and improved.

Author Response

Comments 1: The authors should better discuss how relevant their data actually is. Is vancomycin an antibiotic really widely used in vaginitis treatment? How actually frequent are staphylococcal infections of female urogenital tract?

Response 1: Thank you for the remark! S. aureus is one of key agents causing an aerobic vaginitis (https://pubmed.ncbi.nlm.nih.gov/38182522/#:~:text=Abstract,cycle;%20microbiota;%20vaginal%20tract.) . Hence, studying of how it interacts with vaginal microbiota is important.

Concetrning vancomycin – it can be used intravenously in specific cases such as aerobic vaginitis caused by S. aureus. It is not the first-line drug, however, it is a specific agent.

Comments 2: Why authors actually used Lactobacillus paracasei? This is not a dominant part of vaginal microbiota. L. gasseri seems to be much more relevant object, however, it was not studied as precisely as L. paracasei was.

Response 2: Thank you for the remark! Actually, L. paracasei is not a dominant part of human vaginal microbiota, however, it can be a part of it as well as S. aureus. Also, S. aureus and L. paracasei inhabit human gut, where they can ‘meet’ and form a consortium. Hence, of course, L. gasseri, L. crispatus and other dominative vaginal lactobacilli should be in future an object to investigation, but we decided to start with L. paracasei.

Comments 3: In the section of differential gene expression authors studied S. aureus genes. However, what about L. paracasei? It should be also checked better or at least mentioned in the discussion

Response 3: In L. paracasei we found only vanTG and qacJ genes, however their expression was too unstable. Hence, we decided not to include this part in the text. We added some sentences in the text (lines 1192-1196)

Only two resistance genes were found in L. paracasei AK508 (data not shown). The first is vanTG and the second is qacJ, which is an efflux pump that removes quaternary ammonium compounds. Since their differential expression was unstable in all cases, we decided to focus on S. aureus. However, the presence of vanTG in L. paracasei is one of reasons of its high resistance to vancomycin.

4. Response to Comments on the Quality of English Language

Reviewer 3 Report

Comments and Suggestions for Authors

All comments are in word document. 

Author Response

3. Point-by-point response to Comments and Suggestions for Authors

3.1.Abstract

3.1.1. Length and Focus

Comments 1: The abstract is overly detailed (too long for journal limits). It should be reduced to approximately 200–250 words.

Response 1: Thank you for pointing this out. We revised the abstract to make it more appropriate. However, it was already only 151 words, hence we focused on the key results as you recommended. Also, we shortened the abstract [lines 21-31]

Comments 2: Focus on key results and main conclusions rather than extensive mechanistic descriptions.

Response 2: Agree. We rewrote the abstract (lines 21-31)

“We investigated the effects of vancomycin, estradiol, ethanol, and their combinations on the growth of mono- and binary-species biofilms of Lactobacillus paracasei and Staphylococcus aureus. It was found that, vancomycin at a subinhibitory concentration of 0.001 µg/mL, estradiol, and ethanol acted antagonistically in all cases. This effect was observed across all strains studied. Furthermore, the effects of the active compounds were evident at population, cellular and molecular levels, and were reflected in changes to the count of colony-forming units (CFUs), gene expression and the physiological and biochemical characteristics of cells (e.g. lipid composition of membranes and the extracellular matrix). Therefore, at subinhibitory concentrations of vancomycin in the medium, estradiol can modulate the antibiotic's effect on biofilms, thereby regulating deeply microbial communities.”

Comments 3: Remove experimental details such as solvent effects and carrier adsorption — these belong in the Results section.

Response 3: We did it (see above).

3.1.2. Clarity of the Main Message

Comments 4: The abstract should clearly convey whether estradiol enhances or counteracts vancomycin activity, as the current version is somewhat ambiguous.

Response 4: We tried to write it (lines 29-31).

“Therefore, at subinhibitory concentrations of vancomycin in the medium, estradiol can modulate the antibiotic's effect on biofilms, thereby regulating deeply microbial communities.”

Comments 5: Emphasize the overall biological significance rather than specific gene-level findings

Response 5: Thank you! We tried to do this

3.1.3. Keywords

Comments 6: The list is excessively long (12+ terms). Reduce to 6–8 key terms: biofilms, estradiol, vancomycin, Lactobacillus, Staphylococcus aureus, microbial endocrinology, antibiotic resistance.

Response 6: Thank you. We reduced the key words number (lines 45-48)

biofilms, hormones, estradiol, microbial endocrinology, antibiotics, vancomycin, antibiotic resistance, Lactobacillus, Staphylococcus aureus”

3.2. Introduction

3.2.1. Structure and Readability

Comments 7: The text is dense and overly long, with several redundant statements (e.g., lines 40–47 and 49–55 repeat similar ideas).

Response 7: We removed text in the 49-55 lines (now lines 60-63)

Comments 8: Simplify and shorten sentences to enhance clarity.

Response 8: We focused on those stylistic mistakes and tried to rewrite the text

Comments 9: Avoid double intensifiers like “extensive and intensive investigation” or “remain being a challenge.

Response 9: We focused on those stylistic mistakes and tried to rewrite the text

3.2.2. Rationale

Comments 10: The introduction could more clearly justify why estradiol was chosen, and why its interaction with vancomycin and ethanol is of special interest.

Response 10: we added the part in the text (lines 121-124)

“Due to the strong antibacterial properties of ethanol, which is produced by various vaginal microorganisms [32] and used as an estradiol solvent, it is important to also check the combined effect of ethanol and vancomycin as an additional control.”

Comments 11: Highlight the novelty — for example: “While estradiol’s influence on Gram-negative pathogens is relatively well documented, its role in Gram-positive biofilms and antibiotic modulation remains unexplored.”

Response 11: We added the novelty (lines 92-95)

However, while estradiol’s influence on Gram-negative pathogens is relatively well documented, its role in Gram-positive biofilms and antibiotic modulation, and the mechanisms of estradiol action on microorganisms and their variability remain poorly understood”

3.2.3. Hypothesis and Objective

Comments 12: The research aims at the end (lines 96–100) is appropriate but should be reformulated as a clear, concise hypothesis, e.g.: “We hypothesize that estradiol modulates vancomycin efficacy and biofilm development in mono- and dual-species systems of L. paracasei and S. aureus.”

Response 12: Thank you! We added this in the text (lines 127-129)

3.2.4. Terminology

Comments 13: Replace “binary biofilms” with the more standard term “dual-species biofilms.”

Response 13: We checked the text and changed the terminology

Comments 14: Ensure consistent use of italicized species names throughout.

Response 14: We checked the text and fixed the typos

3.3. Methods

Comments 15: The section is too long and densely written; several subsections (e.g., 2.3–2.5) contain procedural details that could be summarized.

Response 15: We rewrote and shortened those sections in accordance to the recommendations

Comments 16: To improve readability, methods should follow a clear logic: cultures compound preparation biofilm models analytical assays.

Response 16: We rewrote the Methods section completely, now it is organized as it was recommended.

Comments 17: Consider moving lengthy descriptions (e.g., sonication or hybridization conditions) to Supplementary Materials and refer to them briefly in the main text.

Response 17: We removed some long descriptions (i.e. lipids analysis, matrix extraction and RNA extraction) to the Supplementary data.

3.3.1. Replication and statistics

Comments 18: The number of biological and technical replicates is not stated for most assays. Reviewers and editors expect clear statements such as: “Each experiment was performed in triplicate and repeated three times independently.”

Response 18: We added the sentence in the text (Lines 621-622)

“All experiments were performed at least three times independently with two or three technical replicates in each biological repeat”

Comments 19: Indicate the statistical software and methods used to evaluate differences (e.g., ANOVA, Tukey’s test, Student’s t-test).

Response 19: We added the information (lines 625-628)

The nonparametric Mann–Whitney U-test was used to assess statistical significance between groups in microbiological experiments. Student’s t-test was used to assess statistical significance in gene expression experiments. Both tests were included in GraphPad Software.”

3.3.2. Concentrations and experimental ranges

Comments 20: Some concentrations (e.g., vancomycin 0.001–500 μg/mL) are given as a wide range, but it is unclear which specific concentrations were used in each assay. Clarify the rationale for selecting the final working concentration.

Response 20: We added the concentration 0.001 µg/mL in the text (line). And also we added this value in methods descriptions where applicable (lines 650-655)

“This concentration is much lower than used in clinical practice [Kullar et al., 2012 ], and because of such an interesting phenomenon we decided to use this ultralow concen-tration of vancomycin. Also, bBecause S. aureus is an opportunistic pathogen of higher clinical concern, we decided to use a concentration of vancomycin effective against it. Hence, we selected 0.001 µg/mL for subsequent experiments.”

3.3.3. Reproducibility

Comments 21: The description of modified media (e.g., “modified MRS medium”) lacks composition or citation.

Response 21: Actually, the composition of the modified MRS medium was described in lines 189-194

“…modified de Man–Rogosa–Sharpe (MRS) medium with the following composition (g/L): peptone (Dia-M) 20; glucose (Dia-M) 20; yeast extract (Dia-M) 5; sodium acetate (Reachem) 5; ammonium chloride (Reachem) 2; sodium citrate (Reachem) 2; potassium diphosphate (Reachem) 2; Tween 80 (Dia-M) 1; magnesium sulfate (Reachem) 0.1; manganese sulfate (Reachem) 0.05; pH 7.0. “

Comments 22: Details such as gas mixture composition for anaerobic cultivation (“GasPak system”) could be cited or specified (e.g., “BD GasPak EZ, 5% CO₂ atmosphere”).

Response 22: We added a part in the text (lines 260-261)

“Anaerogas sachets provide 14-16% of CO2, 4-6% of H2 and <0.1% of O2.”

3.3.4. FISH and microscopy

Comments 23: Excellent inclusion of probe sequences, but the source and validation of probes should be cited explicitly (e.g., “as previously validated in reference X”).

Response 23: Thank you! We added the sentence (lines 353-354)

The absence of false-positive reactions of probes was validated as described previously [15] and in references [37,38]”

Comments 24: Specify the software version used for Comstat2 or ImageJ plugins.

Response 24: We added the versions (lines 401-403)

“…in ImageJ 1.48v Java 1.6.0_20 (NIH, Bethesda, USA) using the Bio-Formats importer and the Comstat2 plugin (Version 2.1 July 1 2015, University of Copenhagen…”

3.3.5. Lipid and matrix extraction

Comments 25: The lipid analysis section is scientifically rich but described in excessive detail. This could be summarized in one paragraph, with a reference to established protocols (“as described in [reference]”).

Response 25: We removed a part of the methodic to the Supplementary Data

Comments 26: Quantification methods (“densitometry using calibration standards”) should indicate how calibration was performed (linear range, software used).

Response 26: we added a description (lines 532-533)

“Quantitative analysis was carried out by densitometry, using Dens software, version 5.1.0.2 (Lenchrom, Russia), in the linear approximation mode.”

3.4. Specific Suggestions

3.4.1. Terminology

Comments 27: Replace “binary biofilm” with “dual-species biofilm” for terminological consistency.

Response 27: we replaced the terms according to the recommendation

Comments 28: Ensure italicization of all Latin names throughout.

Response 28: we checked the italicization as recommended

3.4.2. Ethical and biosafety statement

Comments 29: While not strictly required for in vitro studies, a brief statement confirming compliance with institutional biosafety standards would strengthen transparency.

Response 29: We added a brief statement into the text (lines 1386-1388).

“The research is conducted in accordance to requirements of the Ethics Committee of the Federal Research Center of Biotechnology of the Russian Academy of Sciences”.

3.4.3. Formatting and conciseness:

Comments 30: Group subsections logically (e.g., 2.4–2.6 could be merged as “Biofilm Models and Quantification”).

Response 30: : Thank you! We rearranged the subsections, now all the biofilm cultivation is in section 2.4 “Cultivation of monospecies and dual-species biofilms” with subsections 2.4.1. “Monospecies biofilms on polytetrafluoroethylene cubes”; 2.4.2. 2.54.2. Monospecies and dual-species biofilms on glass fiber filters

Comments 31: Move tables (such as probe sequences) to Supplementary Material unless explicitly required by the journal.

Response 31: We removed the Table 2 to Supplementary data

3.4.4. Estradiol preparation

Comments 32: The description is correct but could note whether stock solutions were protected from light, as estradiol is light-sensitive — a common reviewer query

Response 32: We added this point in the text (lines 151-152)

“Also, estradiol stock solutions were kept in dark to prevent potential light-mediated degradation.”

3.5. Results.

3.5.1. Specific section comments

3.5.1.1. Section 3.1–3.2

Comments 33: Clear and well-presented; vancomycin concentration effects are logical.

Response 33: Thank you!

Comments 34: The rationale for selecting 0.001 μg/mL vancomycin should be explicitly justified beyond “clinical concern.”

Response 34: : Thank you! We added a part in the text (lines 650-655)

“This concentration is much lower than used in clinical practice [Kullar et al., 2012], and because of such an interesting phenomenon we decided to use this ultralow concentration of vancomycin. Also, because S. aureus is an opportunistic pathogen of higher clinical concern, we decided to use a concentration of vancomycin effective against it. Hence, we selected 0.001 µg/mL for subsequent experiments.”

3.5.1.2. Section 3.3

Comments 35: Good use of CFU quantification, but variability between inoculum levels complicates interpretation

Response 35: Thank you! We suppose that the lower amount of S. aureus cells inoculated in the first experiments led to less viability of S .aureus in binary communities and hence – in lower stability of results. Therefore we decided to compare two ways of inoculation. And, consequently, increase of S. aureus inocula OD resulted in better data. We added some explanation in the text (lines 689-694)

“. This is a logical consequence of the shortage of S. aureus biomass shortage that occurred during the plating of lower OD cultures. Lactobacilli suppressed S. aureus and no pronounced impact was detected. However, while plated in a higher amount, staphylococci were more resilient to the negative effects of L. paracasei and formed a more pronounced dual-species community”

Comments 36: Suggest adding a concise comparison table summarising effects across conditions

Response 36: Thank you! We added Table 2 into the text. Also, we added a text (lines 618-729)

“In summary, Table 2 shows the effects of all active compounds assessed in the initial experiments. Two main conclusions can be drawn. First, estradiol acts as an antagonist to vancomycin. Second, changes were more frequently observed in dual-species biofilms. It is also important to note the absence of correlation between the results of CV staining and CFU counts. This suggests that active compounds may affect the toughness biofilm, resulting in more or less biomass on the PTFE cubes after rinsing procedures..

Table 2. Summary of the results of CFU counting, MTT and CV staining of S. aureus and L. paracasei monospecies and dual-species biofilms. “N/E means” no effect; “N/A” means not applicable; “In” means inhibition; “St” means stimulation; “TIn” means tendency to inhibition; “TSt” means tendency to stimulation. Light-colored cells or colorless cells mean no statistical significance of differences or no difference.

3.5.1.3. Section 3.4

Comments 37: CLSM data are described in excessive numeric detail; summarise with relative changes and representative images

Response 37: Thank you for the remark! We removed some numerics.

Comments 38: The interpretation that estradiol’s effect depends on carrier surface is interesting—this deserves more emphasis.

Response 38: Thank you! We added some thoughts in the text (lines 768-776)

This may potentially be the result of different interactions between hydrophobic and hydrophilic molecules and surfaces. Hydrophilic glass seems to be a worse surface for the physicochemical adhesion of organic molecules, especially hydrophobic ones like estradiol or different components of the cell envelope. Hydrophilic glass also seems to be a worse surface for the formation of a conditioning layer — a layer of organics that covers the surface and accommodates cell adhesion. Estradiol seems to be a cell surface-interacting molecule, and surface nature may be important. As we will show below, estradiol's interaction with the surface is indeed important for its impact on cells.

3.5.1.4. Section 3.5

Comments 39: The section is quite long and complex; consider splitting into two (adsorption effects vs solvent influence).

Response 39: We rewrote this part completely and made new figures.

Comments 40: It is difficult to follow the logic of comparisons between PS, ethanol, and RCM preincubation; a schematic would help.

Response 40: Thank you for the remark! We added a generalizing scheme (fig. 11).

3.5.1.5. Section 3.6

Comments 41: Lipid analysis is impressive but only briefly contextualised. Clarify whether lipid changes correlate with observed biofilm modulation

Response 41: Thank you for the remark! In the system where biofilms were cultivated for matrix isolation and lipid analysis, only minor fluctuations in biomass weight were observed. However, estradiol at a concentration of 2.2×10-10 M decreased the biomass of L. paracasei biofilms in most cases, but increased lipids. Lipid content is in an inverse relationship with biofilm biomass. We added a section to the text (lines 1114-1117)

“It is important to note that estradiol usually inhibited the growth of L. paracasei biofilms at a concentration of 2.2 × 10⁻10 M while increased lipid amounts. Therefore, L. paracasei bio-films potentially become more “saturated” with lipids in the presence of the hormone.”

3.5.1.6. Section 3.7

Comments 42. Antibacterial assay results appear inconsistent; authors appropriately acknowledge variability. However, they could statistically assess reproducibility or correlation with biofilm strength.

Response 42: This is actually a bit unclear because we obtained bacterial lawns in all experiments. In fact, those lawns appeared stable and visually identical, as did the biomass in the matrix experiments. Therefore, there could be other reasons, such as minimal shifts in the atmosphere's composition (e.g., not so well hermitized bags or minimal shifts in the medium's pH). In the future, we plan to perform an antibacterial activity test in parallel with an MTT assay and/or pH measurement. However, as we previously demonstrated [33], L. paracasei does not alter the pH in the presence of estradiol. We added some thoughts to the text (lines 1131-1135)

“Although the bacterial lawns appeared stable and identical in all samples, significant variations in activity were observed. This could result from various factors, including minimal shifts in the atmosphere's composition (e.g., poorly sealed bags), minor alterations in the medium's pH, or minor errors in the medium preparation process. It could also be a consequence of the strain's unstable behavior”

3.5.1.7. Section 3.8

Comments 43: Gene expression data are intriguing but would benefit from clearer visual presentation (heatmap or fold-change plot).

Response 43: We changed the plot (fig. 15).

Comments 44: The statement that “variability was high, and most differences were not statistically significant” should be supported by p-values or error bars.

Response 44: Thank you. We added “p>0.05” in the text (line 1174)

3.5.2. General comments

Comments 45: The English is clear and precise but dense. Shorter sentences would improve readability.

Response 45: We addressed the remark and redacted the text.

Comments 46: Some redundancy (e.g., “estradiol plus vancomycin decreased…” repeated many times).

Response 46: We addressed the remark and redacted the text.

Comments 47: Consider using more summarizing language (e.g., “Overall,” “In contrast,” “Collectively, these results indicate…”) to improve narrative flow.

Response 47: We tried to address this remark, thank you!

3.6. Discussion and conclusions

Comments 48. L752–759: The introductory framing of microbial endocrinology is excellent but somewhat generic; could be shortened by one paragraph to maintain focus on current findings.

Response 48: Thank you! We shortened this paragraph (lines 1205-1217)

The ability of hormones to modulate biofilm growth is now well recognized. Accordingly, the central question has shifted from whether hormones act on bacteria to how they act do so and how their effects can be harnessed Another practical approach is to explore the application use of host hormones as to regulatorse of microbial communities and as enhancers or modulateors of conventional antibiotics. This is particularly relevant for hormones, such as steroids, which are widely used as pharmacological substances. The present study examined the potential of estradiol’s potential to modulate vancomycin susceptibility in selected Gram-positive bacteria.

Comments 49. L770–774: The hypothesis about bacterial receptors sharing 3D features with eukaryotic steroid receptors is intriguing. Suggest citing or discussing known examples (e.g., LuxR solo receptors or steroid-binding regulators in actinobacteria) to anchor this idea.

Response 49: Thank you! We added some thoughts in the text (lines 1228-1236)

“Additionally, Clabaut et al. reported a protein that may potentially act as a steroid sensor in lactobacilli. Clabaut and colleagues discovered a membrane lipid raft-associated SPFH domain-containing protein that shows homology with the eukaryotic estrogen-related receptor gamma (ERR3) and the estradiol-binding protein prohibitin-2 (PHB2) [Clabaut et al., 2021]. Some Mycobacterium species have been described as having steroid-binding cytochromes P450 [Child et al., 2020]. Steroid receptors are typically dimers with DNA-binding domains containing zinc finger motifs (Griekspoor et al., 2007). DNA-binding proteins with a 3D structure similar to this may potentially be steroid (and estradiol)-binding receptors in bacteria, such as different LuxR solo receptors, etc. [da Silva et al., 2015].”

Comments 50. L788–807: Excellent integration of transcriptional data; however, p-values or fold changes should be mentioned explicitly if discussed in this section.

Response 50: we added “a 29-fold increase, p<0.05” in the test (line 1271)

Comments 51. L809–832: The paragraph on adsorption and volatile organics is unique and thought-provoking but somewhat speculative. Recommend marking this as a methodological consideration or moving part of it to the Methods or Supplementary Discussion.

Response 51: Thank you! We placed a part of this text into results section (lines 798-800).

Third, changes in aggregation and matrix toughness could alter resistance to pipetting, resulting in different aggregate sizes and consequently, OD. The most likely explanation is a combination of these factors.

Also, in the discussion section we marked it in accordance to this remare (lines 1289-1296):

Importantly, planktonic cultures in microtiter plates were much more sensitive to active compounds than cultures in other experiments, possibly due to methodological weaknesses. During sampling, standard pipetting can detach variable amounts of loosely adherent biomass. If compounds alter matrix density or aggregation, differences in OD540 may reflect changes in aggregate size and detachment rather than growth alone. Hence, important methodological considerations for future work avoiding the complete removal of the planktonic culture volume by pipetting and creating a diluted suspensions in a new microplate.

Comments 52. L834–845: The instability of antibacterial activity is an important observation; discuss whether estradiol’s effect on extracellular matrix could alter diffusion of bacteriocins.

Response 52: Thank you! We rewrote this section (lines 1317-1326):

“Additionally, potential changes in the composition of the extracellular matrix (particular-ly its lipid compartment, as revealed in this study) may explain the instability of antibac-terial activity. However, the biomass taken from the Petri dishes in matrix experiments appeared stable. Nevertheless, the most susceptible indicators, including micrococci, ky-tococci, and Pseudomonas chlororaphis, were inhibited by L. paracasei in all six repeats. Pre-viously, it was suggested that the biofilm life form allowed Lactobacillus to exhibit more pronounced antimicrobial activity. However, it has been reported that different Lactobacil-lus species exhibit varying degrees of dependence on their antimicrobial activity when switching from a planktonic to a biofilm state [62]. Thus, L. paracasei may also exhibit such agile behavior.”

Comments 53. L846–855: The summary paragraph is solid, but the final sentence is incomplete (“…applications in microbiology and”). It should be edited for completeness and clarity.

Response 53: Thank you! We completed this sentence (lines 1343-1344)

These findings underscore estradiol’s capacity to influence multispecies biofilms and support further investigation into its molecular targets and potential applications in microbiology and medicine

Comments 54. Conclusions (L857–867): Well written, concise, and logically aligned with the aims. Still, the claim that estradiol is a “potent regulator” could be softened unless quantitative measures of potency relative to other hormones are available.

Response 54: We softened the sentence (lines 1346-1348):

“Estradiol can regulate the growth and metabolism of Staphylococcus aureus and Lactobacillus paracasei biofilms”

Comments 55. Scientific English is clear and precise, though sentences are often long (25–40 words). Shorter, more direct phrasing would improve readability.

Response 55: Thank you! We edited the text.

Comments 56. Avoid repetition of “antagonistic”, “stimulatory”, “inhibition”, etc., by occasionally using synonyms or summarizing patterns.

Response 56: We tried to address this remark, thank you!

Comments 57. The paragraph transitions could be smoother — e.g., using connectors like “Moreover”, “Conversely”, “Taken together”, etc.

Response 57 : We tried to address this remark, thank you!

4. Response to Comments on the Quality of English Language

Point 1: Several sentences are long and complex, reducing readability (e.g., lines 22–29).

Response 1: Thank you! We tried to rewrite the text.

Point2: Use shorter, clearer sentences and standard scientific phrasing. Example: “Estradiol modulated vancomycin activity in a species-dependent manner, altering gene expression and biofilm lipid composition.”

Response 2: thank you! We corrected the text

Point3: The manuscript would benefit from professional English editing to correct occasional grammatical errors and improve fluency.

Response 3: Thank you! We corrected the text

Point4: Some sentences are syntactically overloaded and could be divided into two or more simple clauses.

Response 4: thank you! We corrected the text

Point5: Use formal, concise scientific English throughout (avoid conversational or redundant expressions).

Response 5: thank you! We corrected the text

5. Additional clarifications

[Here, mention any other clarifications you would like to provide to the journal editor/reviewer.]

Round 2

Reviewer 1 Report

Comments and Suggestions for Authors

It can be accepted in its current form.